# Detection of nitrous acid in the atmospheric simulation chamber SAPHIR using open-path incoherent broadband cavity-enhanced absorption spectroscopy and extractive long-path absorption photometry

Sophie Dixneuf[1§], Albert A. Ruth[1*], (the late) Rolf Häseler[2┼], (the late) Theo Brauers[2╫],
Franz Rohrer[2], and Hans-Peter Dorn[2]

[1]Department of Physics & Environmental Research Institute, University College Cork, Cork, Ireland.
[2]Institut für Energie und Klimaforschung, IEK-8: Troposphäre, Forschungszentrum Jülich GmbH, 52425 Jülich, Germany.

[§]Current address: Bioaster Technology Research Institute - Bioassays, Microsystems & Optics Engineering Unit, 40 Avenue Tony Garnier, 69007 Lyon, France

*Correspondence to*: Albert A. Ruth (a.ruth@ucc.ie)

┼ Rolf Häseler - passed away on 25 July 2017.
╫ Theodor Brauers - passed away on 21 February 2014.

**Abstract.** An instrument based on 20 m open-path incoherent broadband cavity-enhanced absorption spectroscopy (IBBCEAS) was established at the Jülich SAPHIR chamber in Spring 2011. The setup was optimized for the detection of HONO and $NO_2$ in the near UV region 352-386 nm, utilizing a bright hot-spot Xe-arc lamp and a UV-enhanced CCD detector. A $2\sigma$ detection limit of 26 pptv for HONO and 76 pptv for $NO_2$ was achieved for an integration time of 1 min. Methacrolein (MACR) was also detected at mixing ratios below 5 ppbv with an estimated $2\sigma$ detection limit of 340 pptv for the same integration time. The IBBCEAS instrument's performance for HONO and $NO_2$ detection was compared to that of extractive wet techniques, long-path absorption photometry (LOPAP) and chemiluminescence spectrometry (CLS) $NO_X$ detection, respectively. For the combined data sets an overall good agreement for both trend and absolute mixing ratios was observed between IBBCEAS and these established instruments at SAPHIR. Correlation coefficients r for HONO range from 0.930 to 0.994 and for $NO_2$ from 0.937 to 0.992. For the single measurement of MACR r = 0.981 is found in comparison to proton transfer reaction – mass spectrometry (PTRMS).

## 1 Introduction

Photolysis of nitrous acid (HONO + hν (λ < 400 nm) → OH + NO) leads to the formation of the most important daytime oxidant in the atmosphere, the hydroxyl radical (OH). Thus HONO indirectly affects the oxidative potential of the troposphere and strongly influences degradation mechanisms of a vast variety of natural and urban pollutants. The mechanisms of HONO formation in the troposphere are still not fully understood (Calvert et al. 1994, Finlayson-Pitts et al. 2003, Ramazan et al. 2004, Liu et al. 2019). Many atmospheric studies revealed elevated HONO mixing ratios during daytime under specific conditions

that cannot be fully explained (Staffelbach et al. 1997, Zhou et al. 2002a, Zhou et al. 2003, Kleffmann et al. 2003, Vogel et al. 2003, Kleffmann et al. 2005, Acker et al. 2006, Spataro and Ianniello, 2014). Although there appears to be general agreement that heterogeneous $NO_2$ chemistry is one of the most important sources of HONO (Harrison et al. 1994, Reisinger 2000), modelled HONO mixing ratios are often significantly below observed values (Vogel et al. 2003, Lammel and Cape 1996). However, other sources have also been suggested (Stemmler et al. 2006, Li et al. 2012, 2014). Since the photochemistry of

HONO is closely connected to that of $NO_2$ (Johnston et al. 1974, Aumont et al. 2003, Bröske et al. 2003, Ramazan et al. 2004), the in situ measurement of time-dependent HONO and $NO_2$ mixing ratios by monitoring both species simultaneously is particularly interesting to elucidate the natural formation processes of HONO (Kleffmann 2003). One reason for the indeterminate formation and atmospheric role of HONO is the challenge to accurately and reliably quantify this species.

A direct spectroscopic way to detect HONO is through its electronic absorption in the near UV between 320 and 390 nm (Stutz

et al. 2000), or through its IR active vibrational modes (Barney et al. 2000), e.g. in the Q-branches of trans-HONO at 1263 cm$^{-1}$ ($ν_3$) or 790 cm$^{-1}$ ($ν_4$). The cross-sections in both cases are approximately between ~2 and $6 \times 10^{-19}$ cm$^2$/molecule. The UV region has been extensively exploited in differential optical absorption spectroscopy (DOAS) (Febo et al. 1996, Alicke et al. 2003, Stutz et al. 2010), but more recently also using cavity enhanced methodologies (Wang and Zhang 2000, Djehiche et al. 2011, Jain et al. 2011), especially incoherent broadband cavity enhanced absorption spectroscopy (IBBCEAS) (Gherman et

al. 2008, Hoch et al. 2012, Wu et al. 2012, Donaldson et al. 2014, Scharko et al. 2014, Min et al. 2016, Yi et al. 2016, Nakashima and Sadanaga 2017, Duan et al. 2018, Jordan and Osthoff 2020, Tang et al. 2020, and Yi et al. 2021). The mid-IR was targeted using Fourier transform (Hanst et al. 1982) and tuneable diode laser spectroscopy (TDLAS) (Schiller et al. 2001). Laser-induced fluorescence (LIF) can also be used for sensitive HONO detection, but only through the emission of OH radicals that are formed after HONO photolysis (Rodgers and Davis 1989, Liao et al. 2006, Bottorff et al. 2021).

Most non-spectroscopic (indirect) detection methods are chemical in nature. Typical approaches comprise dry (Ferm and Sjödin 1985), and wetted effluent diffusion (Simon and Dasgupta 1995, Acker et al. 2006), or rotated (Oms et al. 1996, Spindler et al. 2003) denuders, HPLC/fluorescence methods (Huang et al. 2002, Takenaka et al. 2004, Beine et al. 2005), or long path absorption photometry (LOPAP) (Heland et al. 2001, Kleffmann et al. 2002). The corresponding instrumentation is generally more sensitive than spectroscopic methods, but at the same time also more susceptible to chemical interferences that can affect

the selectivity and quantification of HONO (Beine et al. 2005, Kleffmann et al. 2006). In most of the chemical methods HONO is sampled on humid or liquid surfaces where heterogeneous chemistry can affect HONO mixing ratios in the presence of

specific interfering chemicals (Ferm et al. 1985, Spindler et al. 2003, Zhou et al. 2003, Gherman et al. 2008). Moreover, photolytic or heterogeneous formation of HONO in sampling lines can cause unreliable results in in situ measurements (Zhou et al. 2002a, 2002b, Rohrer et al. 2005). Therefore for studies aiming at HONO detection it is essential to compare the performance of instruments that use genuinely different approaches (Kleffmann 2007). For example, validation studies of denuder based approaches against DOAS measurements illustrated that significant discrepancies exist for daylight/illuminated conditions (Kleffmann 2007). DOAS was also compared with the LOPAP approach and several reasons for systematic differences in the acquired data were identified (Kleffmann et al. 2006).

A long open-path broad-band cavity enhanced absorption instrument was set up at the SAPHIR[1] chamber at Forschungszentrum Jülich (Germany) in 2011, utilizing an incoherent short-arc Xe lamp as a light source (cf. Fiedler et al. 2003, Varma et al. 2009). Results of measurements to characterize its performance in the near UV through the in situ detection of HONO and $NO_2$, as well as methacrolein (MACR, also called methacrylaldehyde or isobutenal) are presented here. MACR is of interest as "interfering species" which is naturally formed together with methyl vinyl ketone (MVK, also called butenone) (Pierotti et al. 1990) as secondary atmospheric product after the reaction of isoprene with OH (Fehsenfeld et al. 1992). Both MACR and MVK are important factors in the oxidation chemistry of biogenically emitted species such as isoprene, and can lead to the formation of ozone and/or secondary organic aerosol (Wennberg et al. 2018). MACR also has primary emission sources in urban environments. It is generated in combustion processes and industrial activities (Destaillats et al. 2002) and it has even been detected in indoor air from cigarette smoke (Sleiman et al. 2104). Typical natural mixing ratios of MACR of tens of pptv to < 3 ppbv have been observed e.g. above forest canopies in western Alabama (Montzka et al. 1990) and in airborne monitoring campaigns in Surinam using proton transfer reaction – mass spectrometry (PTRMS, Warneke et al. 2001). In urban environments mixing ratios of e.g. tens of pptv to ~0.33 ppbv have been reported for a 4 hr period at the San Francisco bay bridge toll plaza using high-resolution gas chromatography/ion trap mass spectrometry (Destaillats et al. 2002).

Similar to the recent work by Yi et al. 2021 the objectives of our work in 2011 were (i) to assess the instrument's performance concerning HONO detection and to validate it with the LOPAP system available at SAPHIR, and (ii) to compare its performance concerning $NO_2$ detection with the local SAPHIR chemiluminescence spectrometry (CLS) $NO_X$ detector.

Section 2 outlines details on the experimental setup at SAPHIR in Jülich. Section 3 shows measurements of HONO, $NO_2$ and MACR mixing ratio using the IBBCEAS instrument. The data are respectively compared with results from different instruments at SAPHIR: (a) a LOPAP system (HONO), (b) a standardized chemiluminescence detector ($NO_2$), and (c) a proton-transfer reaction mass spectrometer (PTRMS) (MACR). Correlation plots between individual measurements will be discussed for three measurement days in Summer and Autumn 2011. The experiments presented here supplement a campaign on the "Formal Intercomparison of Observations of Nitrous Acid" (FIONA) (Rodenas et al. 2013), where instruments for the quantification of HONO were compared at the EUPHORE simulation chambers in Valencia (Spain) in May 2009.

---

[1] SAPHIR = **S**imulation of **A**tmospheric **PH**otochemistry **I**n a large **R**eaction chamber)

## 2 Experiment

IBBCEAS has been used for the detection of a variety of target species in different wavelength regions in the laboratory (Langridge et al. 2006, Washenfelder et al. 2008, Dixneuf et al. 2009, Thalman and Volkamer 2010, Wu et al. 2009, Nakashima and Sadanaga 2017, Duan et al. 2018) and in outdoor environments (Bitter et al. 2005, Saiz-Lopez et al. 2006, Leigh et al. 2010, Wu et al. 2012). The high sensitivity and spatial resolution of open-path IBBCEAS make this approach particularly attractive for applications in atmospheric simulation chambers (Varma et al. 2009, Fuchs et al. 2010, Chen et al. 2011, Ashu-Ayem et al. 2012, Hoch et al. 2012, Dorn et al. 2013, Rodenas et al. 2013). Thus an open-path IBBCEAS instrument was developed at the Jülich SAPHIR chamber, a facility designed for the simulation of tropospheric scenarios at realistically low mixing ratios of relevant trace species (see e.g. Rohrer et al. 2005). The SAPHIR chamber is an ideal testbed for open-path IBBCEAS due to the available comprehensive suite of standard detection methods that can be used to assess and validate the data taken with IBBCEAS. For the experiments presented here the IBBCEAS instrument was optimized for the near-UV detection of HONO (352-386 nm), whose identification was based on three prominent absorption bands peaking at $\approx$354, $\approx$368 and $\approx$384 nm (Stutz et al. 2000). The spectral region covered also enabled the simultaneous detection of $NO_2$ and methacrolein. The experimental design of the IBBCEAS instrument was similar to the one described in Refs. (Varma et al. 2009, Dorn et al. 2013, Varma et al. 2013). It comprised a transmitter and a receiver unit, each housing one of the cavity mirrors. A sketch of the optical setup and details on experimental components are shown in Figures S1-S3 in the supplementary material. The transmitter unit was mounted on a concrete platform at the north end of SAPHIR (facing south). The receiver unit was located at the south end of SAPHIR (facing north), in order to minimize exposure of the detector to potential daytime stray light. The concave dielectric mirrors (radius of curvature: –21 m, reflectivity ~0.999, diameter = 40 mm, Layertec GmbH) inside the mechanically stable units formed an optical cavity with a mirror separation of ~20.4 m. Each unit was temperature stabilized with an air-to-air thermoelectric assembly (Laird Technologies). Vertical metal plates with optical ports (Figure S3) were rigidly mounted to the concrete support and formed the interface between the instrument's units and the chamber. Pipes connected to the two units extended into the SAPHIR chamber through the optical ports. Due to the use of the vertical plates (sealed by o-rings) the mechanical stress of the chamber's teflon wall was not imposed directly on the two optical units, which essentially decoupled the instrument from the chamber wall and improved its long-term stability. Each unit had two ports: one pipe with 50 mm diameter and another one with 25 mm. In each unit the wider pipe was aligned along the optical axis of the cavity reducing the contribution of stray light to the measured signal in the receiver unit. The pipes were sealed with a high reflectivity mirror at one end and purged with zero air at a flow rate of 1.7 $dm^3$ $min^{-1}$ during measurements to protect the mirror. The narrower pipe, which was sealed off during normal use of the instrument, was only used as a port for a green alignment laser (see Figure S1). The main light source was a short-arc Xe lamp running in "hot-spot" mode. In this mode, a small plasma spot (~150 μm diameter) with a spectral radiance of 18 W $cm^{-2}$ $sr^{-1}$ $nm^{-1}$ at 400 nm was formed close to the cathode surface, improving the imaging properties of the discharge in comparison to conventional diffuse arcs. Small random jumps of the arc position, however, required an active stabilization of the beam direction by means of a quadrant detector.

When the spot "jumped" to a specific quadrant an actively controlled piezo-driven Al-mirror was used to minimise the resulting beam deviation from the optical axis of the CEAS instrument. A telescope imaged the incoherent light into the optically stable cavity whose mirrors were aligned by two remote-controlled high resolution positioning motors per mirror. The light transmitted by the cavity was collected by a UV-enhanced Al-mirror and focused onto the aperture of a circular-to-rectangular

fiber bundle which guided the light onto the entrance slit (25 μm) of a (Shamrock 303i) spectrograph (f = 303mm, F/4) supplied with a 1200 grooves/mm holographic grating. The light transmitted by the spectrograph was imaged onto a CCD detector cooled to –65°C. Light outside the high reflectivity range of the cavity mirrors was optically filtered by means of a band-pass filter to avoid excessive scattering into the spectrometer and the potential saturation of the detector. The wavelength range from 352 to 386 nm was covered with a spectral resolution of ~0.24 nm. A sample of transmission spectra can be found in the

supplementary material (Figure S4).

## 2.1 Measurement Procedure

After overnight flushing of the chamber with dry synthetic air (also referred to as "zero air" in this publication) the cavity transmission, $I_0(\lambda)$, was measured every morning in the dark chamber before experiments commenced. Typically 300 cavity transmission spectra with an individual acquisition time of 200 ms were accumulated during experiments, yielding a time

resolution of 1 min per measured spectrum $I(\lambda)$. The extinction coefficient $\varepsilon(\lambda)$ [cm$^{-1}$] was calculated according to (the left part of) Eq. (1):

$$\varepsilon(\lambda) = \frac{1 - R_{\text{eff}}(\lambda)}{d}\left(\frac{I_0(\lambda)}{I(\lambda)} - 1\right) = a_0 + a_1\lambda + a_2\lambda^2 + a_3\lambda^3 + a_4\lambda^4 + \sum_{i=1}^{N} n_i\sigma_i(\lambda) \tag{1}$$

where $R_{\text{eff}}$ represents the wavelength dependent effective reflectivity of the cavity mirrors and $d = 19.57$ m the interaction path length per pass in the cavity ($d$ equals the mirror separation minus the length corresponding to the mirror purge volume, see

supplementary material, Figure S1). $(1 - R_{\text{eff}})$ comprises all effective optical losses of the cavity when filled with zero air after purging the chamber overnight; i.e. mirror reflectivity losses, Rayleigh (and potentially) Mie scattering losses, diffraction losses. The spatial average of the number density $n_i$ [molecule cm$^{-3}$] of the absorbing species i (= HONO, NO$_2$ and MACR) was retrieved by fitting the function on the right hand side in Eq. (1) to the measured extinction coefficient. $\sigma_i$ [cm$^2$ molecule$^{-1}$] represent the absorption cross-sections of species i. The cross-sections used for HONO, NO$_2$, and MACR were taken from

the references Stutz et al. 2000, Mérienne et al. 1995, and Meller et al. 1990, respectively. Several reference spectra from the literature (Bogumil et al. 2003, Burrows et al. 1998, Harder et al. 1997, Mérienne et al. 1995, Vandaele et al. 2002, Voigt et al. 2002) were tested to fit the NO$_2$ absorption spectra in the 352-386 nm spectral range. The reference data yielding the smallest residuals in measurements with only NO$_2$ were chosen as reference cross-sections in measurements with all other gas mixtures – this turned out to be Mérienne et al. (1995). A similar approach was taken for HONO with reference data from

Refs. Bongartz et al. 1994, Brust et al. 2000, and Stutz et al. (2000). The data from Stutz delivered the smallest least square discrepancies in the fit. All cross-sections were always converted to the spectral resolution of the spectrometer using home-

made Gaussian convolution software, written in Fortran. The five fit parameters $a_j$ (j = 0...4) in Eq. (1) belong to a fourth-order polynomial, accounting for unspecified additional losses, such as background featureless absorption, or Rayleigh and Mie scattering that may become relevant over time. The fitting procedure was based on least squares minimization using a singular value decomposition (SVD) procedure (Press et al. 1986, Varma et al. 2009) in order to eliminate biases of the fit due to parameter correlations. During a first test run of the SVD approach the wavelength of all cross-section reference spectra were individually shifted to further minimize the least square sum, and the optimized wavelengths were subsequently used in SVD analyses. For all reference spectra the typical shift was ≈0.10 ± 0.05 nm and thus within the spectral resolution of the measurement. The absolute wavelength was calibrated with a low-pressure neon pen ray lamp (see also Figure S9). An example of an IBBCEAS extinction spectrum and the corresponding fit of eq (1) to the measured data are shown in Figure 1 (uppermost panel). We will further discuss this figure as well as reflectivity calibration aspects in section 4.

**Figure 1**

## 2.2 The LOPAP Instrument

The LOPAP instrument used in this study has been described in detail by Häseler et al. 2009 and Li et al. 2014 in the context of several campaigns on airborne detection of HONO aboard a Zeppelin airship. Air is extracted through two sampling coils in series where HONO is stripped into the liquid phase. In both coils the air is exposed to equal flows of a solution (0.06 M sulphanilamide in 1 M HCl solution) in which HONO almost instantaneously reacts to forms a diazonium salt. While the first coil removes HONO nearly quantitatively from the gas phase, only a fraction of other chemically interfering species are scrubbed. The second coil, however, samples that fraction of interfering species but only the remaining small amount of HONO. Using the difference of the signals derived from the two coils enables the influence of interfering species to be accounted for. The air is then separated from the liquid and the solutions are separately transferred into two mixing volumes, where a 0.8 mM solution of N-(1-naphthyl)ethylenediamine-dihydrochloride is added to generate an azodye. The azodye's concentration is then determined by its optical absorption to determine HONO mixing ratios. The absorption cells for both channels consist of long length teflon tubing, acting as a liquid core waveguide (LCW). Visible light is sent through the tubing and detected by two small spectrometers.

The entire instrument was housed in a compact 19" rack (56(w) × 60(d) × 100(h) cm$^3$) consisting of two "chemistry" units and one "electronics/detection unit". In order to avoid sampling artefacts in inlet lines the sampling unit was straightforwardly mounted inside SAHPIR about 0.3 m from the chamber wall at the north end and 1 m above the floor. The instrument's sampling frequency, time resolution, 3σ detection limit, 1σ precision, and accuracy of the instrument was 0.33 Hz, 4-5 min, 10 pptv, 3 pptv, and 12%, respectively. The measurement range of the instrument can in principle be varied by the length of the absorption tubes and by the use of different absorption wavelengths for the evaluation. In this study we used an optical path length of 2.9 m.

## 2.3 Chemiluminescence Spectrometry NO$_X$ Detector

The NO$_X$ detector was located in a container underneath the chamber from where gas mixtures were sampled at a flow rate of 1 dm$^3$ min$^{-1}$ through a teflon tube of ca. 6 m length (internal diameter 4 mm) corresponding to an approximate residence time of 1 s. NO$_2$ was converted to NO by a pulsed LED photolytic converter at 395±8 nm (Droplet Measurement Technologies, BLC) in a volume of 17 ml with a conversion efficiency of approximately 50%. The LED in the converter was consecutively switched on and off to alternately determine NO and NO$_X$ concentrations. NO was detected by a customized CLS detector (Eco Physics TR 780 (Rohrer and Brüning 1992, Fuchs et al. 2010). NO$_2$ mixing ratios were calculated using an interpolated value between two subsequent NO$_X$ measurements at a time when NO mixing ratio were measured. The instrument was calibrated using NO standard gas mixtures (2 ppmv NO in N$_2$, BOC-Linde) and gas phase titration with O$_3$ for NO$_2$.

The fact that besides NO$_2$ also HONO is photolysed at 395 nm to yield NO was accounted for by determining the corresponding NO yield from HONO numerically from the spectrum of the LEDs. The HONO photolysis contribution to NO is less than 5% compared to that of NO$_2$. The 1$\sigma$ accuracy of the chemiluminescence detector for NO$_2$ was determined to be ±7%, based on the uncertainty of ±5% of the NO standard used for the calibration and a ±5% uncertainty for the NO$_2$ conversion efficiency. The known interference of 5% towards HONO is not corrected in the final dataset and not included in this accuracy estimate.

## 2.4 Proton Transfer Reaction – Mass Spectrometry (PTRMS)

PTRMS was utilized to monitor methacrolein in the presence of HONO and NO$_2$. Generally the PTRMS technique relies on soft chemical ionization to detect gaseous trace components. The target species are converted to product ions through the transfer of a proton from the reagent ion, H$_3$O$^+$. The trace gases (X) are identified through the mass of the product ions usually being the protonated molecular mass (XH$^+$): H$_3$O$^+$ + X $\rightarrow$ XH$^+$ + H$_2$O. The PTRMS instruments applied here was a quadrupole mass spectrometer system (PTR-Quad-MSThe system features a switchable reagent ion source with H$_3$O$^+$, NO$^+$ and O$_2^+$ as precursor ions for the measurement and identification of a number of trace gases. Details on the PTRMS instrument were published earlier by Wisthaler et al. 2008.

## 3 Results

Measurements of time-dependent mixing ratios of HONO, NO$_2$ and MACR using open-path IBBCEAS were taken during Summer and Autumn 2011, and compared with those utilizing long-path absorption photometry (LOPAP), chemiluminescence spectrometry (CLS) and proton transfer reaction mass spectrometry (PTRMS), respectively. The performance intercomparison study is exemplified on basis of measurements on the 11[th] of July and on the 5[th] & 6[th] of October 2011, when different photo-chemical scenarios were simulated. The measurements in July were carried out as part of a Jülich internal photochemistry campaign (6[th] June to 15[th] July 2011), whose goal was to study the oxidation of isoprene (H$_2$C=C(CH$_3$)-CH=CH$_2$), methacrolein CH$_2$=C(CH$_3$)–CHO or methyl vinyl ketone (CH$_3$-C(O)-CH=CH$_2$) by hydroxyl (OH) radicals at low NO$_X$ mixing

ratios (Nehr et al. 2014, Fuchs et al. 2014). In contrast, the measurements in October were specifically designed for a comparison between LOPAP and IBBCEAS under well controlled low concentration conditions with no obvious potential chemical interferences disturbing the LOPAP instrument. Generally, after cleaning and humidifying the SAPHIR chamber HONO formation by unknown photo-induced reactions on the Teflon chamber walls and degradation was studied in experiments always including light-induced and dark reactions of HONO formation or destruction – the experimental protocols concerning changes in chamber conditions are given in the captions of Figure 2-4. The results obtained on the three days will be outlined in chronological order. The performance of the open-path IBBCEAS instrument will subsequently be discussed in the context of the different measurement condition and atmospheric scenarios together with that of the Jülich LOPAP instrument.

<p style="text-align:center; color:red;">**Figure 2(a) and 2(b)**</p>

## 3.1 Measurements on 11 July 2011

Figure 2(a) and 2(b) summarizes the time-dependent measurements of mixing ratios of HONO, $NO_2$ and MACR as determined by IBBCEAS (black symbols), LOPAP (red), CLS (blue) and PTRMS (green). The same color code is also used in all remaining Figures. The vertical arrow in Figure 2(a) indicates the times when the cavity transmission through the clean chamber, $I_0(\lambda)$, was measured for ca. 10 min. 11 July is the only day for which the build-up of HONO was monitored during daylight conditions. The HONO mixing ratio increased after humidification of the bright chamber (for 48 min) and decreased subsequently when $O_3$ (~40 ppbv) was introduced. At 9:15 hrs there is a marked but unexplained change in the data of HONO mixing ratios as measured by IBBCEAS in comparison to LOPAP. The increase in the noise of the IBBCEAS data occurred ca. 15 minutes before the addition of CO to the chamber at 9:30 hrs, which is neither expected to influence the HONO chemistry nor the data retrieval, even at high CO (~750 pbbv) concentrations (see Figure 2(a)). Presently there is no obvious explanation for this behavior in the IBBCEAS measurement. Likewise the return to dark conditions at 15:33 hrs led initially to an unexpected increase of the HONO mixing ratio as recorded by the LOPAP instrument, but it was also observed by the IBBCEAS measurement. This observation was also made in other campaigns and will be briefly discussed in section 4.2. $NO_2$ mixing ratios, as measured by CLS, increased gradually during humidification of the bright chamber and increased sharply when $O_3$ was introduced, followed by a more gradual increase until MACR was added to the mixture at 11:51 hrs. The jump of the $NO_2$ mixing ratio at 07:30 shows the effect when $O_3$ was added actively to the chamber. The vast majority of NO was oxidised to $NO_2$ at that time (see Figure 2(a)). Both CLS and CEAS show this effect. The observed temporary decrease of the $NO_2$ concentration after the addition of MACR can be explained by a reaction sequence following the formation of OH from HONO: The primary oxidation step of the reaction of OH with MACR forms the peroxy methacryloyl radical ($CH_2=C(CH_3)-C(O)OO$), part of which further reacts with $NO_2$ to form MPAN (peroxy methacryloyl nitrate, $CH_2=C(CH_3)-C(O)OONO_2$). MPAN is thermally unstable with a small thermal decomposition rate of ~ $4.6\times10^{-4}$ s$^{-1}$ at 298 K (Roberts and Bertman 1992); the temperature inside the chamber after the addition of MACR was in excess of 300 K (see Figure S8). Thus a thermal

<p style="text-align:center;">8</p>

equilibrium is established from which $NO_2$ is reformed after the initial amount of MACR has been consumed (Fuchs et al. 2014).

Concentrations of $NO_2$ and MACR both appeared to stagnate when the roof was closed (at 15:33 hrs) due to the absence of light-driven photo-chemistry. Finally the purging of the chamber with synthetic air removed all trace gases from the chamber. The experiments in October were designed to enable accurate mirror reflectivity calibration and intercomparison of LOPAP

and IBBCEAS under unperturbed condition at sub-ppbv mixing ratios of HONO. The measurements shown in Figures 2-4 demonstrate that the long-cavity IBBCEAS instrument at SAPHIR is capable of detecting HONO at pptv levels without difficulty. The ability of the IBBCEAS instrument to measure MACR selectively and with high sensitivity is a useful addition for the evaluation of photochemical experiments of the radical driven oxidation of isoprene since the detection of the jointly formed reaction products MACR and MVK (methyl vinyl ketone) by mass spectrometric techniques using $H_3O^+$ as reactive

ion can only determine the sum of MACR and MVK in the gas phase.

### 3.2 Measurements on 5 October 2011

Figure 3 summarizes the measurement on 5 October 2011. After overnight flushing in the morning of 5 October, $NO_2$ was added to the chamber in steps of 250 pptv (at 6:30 hrs and 6:50 hrs), 500 pptv (at 7:10 and 7:30), and 1 ppbv (at 7:50 hrs and 8:10 hrs). After subsequent humidification for 44 min (starting at 8:41 hrs) and the exposure to daylight at 9:27 hrs, HONO

was formed at levels of up to 400 pptv. Finally, after closing the chamber roof (14:23 hrs), HONO was removed by flushing with zero air (starting at 15:31 hrs). The HONO data from the IBBCEAS and LOPAP setups, and the $NO_2$ data from the IBBCEAS and CLS instruments showed good agreement on that day (Figure 3).

**Figure 3**

### 3.3 Measurements on 6 October 2011

Figure 4 summarizes the measurement on 6 October 2011. It is known that the photo-enhanced formation of HONO in the SAPHIR chamber can be described by an empirical function depending on relative humidity, solar irradiation and temperature (Rohrer et al. 2005). Minimal HONO production was thus achieved by only humidifying the chamber for 42 min after overnight flushing with zero air. At 12:18 hrs the chamber was exposed to daylight leading to the gradual formation based on vestiges of $NO_X$ on the chamber wall. The variation of temperature was limited to the natural variability. After a gradual increase the

HONO mixing ratios leveled off at ca. 250 pptv before the chamber was eventually closed at 16:24 hrs. The correlation between the data obtained with IBBCEAS and LOPAP, as represented by the correlation coefficient of r = 0.970 (Table 1), is still rather satisfactory at these low levels. For $NO_2$ however, the correlation is less pronounced on this day as there appears to be a gradual drift between data from the IBBCEAS and CLS instruments as evident through a discrepancy in the slope of ~38%. Even though this is the largest discrepancy observed (see Table 1) the data are all taken at $NO_2$ mixing ratios below 300 pptv, which

is close to the $2\sigma$ detection limit of 76 pptv. Approximately one third of the data points are below that limit (see Figure 4), which puts these values into perspective.

## 4 Discussion of Instruments' Performances

### 4.1 IBBCEAS Instrument

The main experimental uncertainties determining the quality of the IBBCEAS data reported here are systematic; they are: (a) the stability of the light source, (b) the in situ calibration of the mirror reflectivity, (c) the data analysis and concentration retrieval approach, and (d) the unspecified mechanical instabilities of the setup such as potential thermal drifts or deficiencies of the opto-mechanical components (Ruth et al. 2014). The latter become more critical with increasing cavity length since small changes in the optical alignment have a more severe influence on the instrument's performance; see Varma et al. 2009.

**4.1.1 Lamp Stability**

Short-term intensity fluctuations are due to random spatial variations of the hot spot plasma arc (small arc jumps), whose effects on the optical alignment were minimised by an active quadrant detector control unit (Varma et al. 2009). As long as intensity fluctuations of the lamp do not show any spectral dependence in the wavelength range of interest, the resulting baseline changes can in principle be accounted for in the fit of eq (1) to the measured data. If the fluctuations are however

accompanied by random spectral variation, the retrieval by SVD becomes increasingly difficult. Spatial jumps that were at the compensation limit of the quadrant detector were also noted during experiments leading to uncertainties in the baselines. One of these events may have occurred e.g. on 11 July at ~9:30 hrs. Difficulties in the mixing ratio retrieval due to lamp instabilities occurred sporadically during the measurements in the Summer, but became more frequent and for longer time periods after continued use of the lamp in the Autumn. For instance the measurement on 6 Oct (Figure 4) was affected by lamp stability

issues, which is a sign of electrode aging and the main cause for the hot spot plasma arc to wander into a domain where the stabilization system is unable to fully compensate for the spatial displacement.

An unexplained change in performance started approximately 20 min before the addition of 750 ppbv of CO on 11 July at ca. 9:10 hrs (Figure 2(a)). At that time the quality of the IBBCEAS data started to worsen, while the LOPAP performance remained largely unaffected (see Figure 2). A small increase in HONO mixing ratios was subsequently observed in comparison to the

LOPAP and the noise of the IBBCEAS data was increased by roughly a factor of ~2. There are no obvious reasons for this observed behavior from a (photo)chemical point of view, since at the time the gas mixture was not altered. In open path cavity setups similar behavior can in principle be caused through increased scattering due to particle formation (Varma et al. 2009), but measurements of the particle number concentration showed no significant change on this occasion. Therefore one conceivable explanation may be that the hot spot arc in the Xe lamp may have moved to a steady location on the electrode

where the control unit was at its performance limit to keep the arc steady. As the quadrant detector correction signal was not recorded, this is merely a tentative explanation of the sudden change of performance. Although there was an addition of CO a short time later it is not plausible that potential impurities in the CO (purity >99.9%) gas (or chemical reactions of same) might

be the cause for this observation. LED-based IBBCEAS is generally less prone to sudden changes (Gherman et al. 2008) due to the higher stability of the light source, although slow drifts are still possible (Fouqueau et al. 2020).

## 4.1.2 Mirror Reflectivity and Calibration Aspects

Retrieval of accurate mixing ratios by IBBCEAS requires the effective mirror reflectivity $R_{eff}$ to be known accurately as a function of wavelength. Calibration measurements were performed a few times over the course of the measurements presented here by introducing a known amount of $NO_2$ into the dry SAPHIR chamber shortly after the cavity transmission of the clean and dry cavity, $I_0(\lambda)$, had been recorded (Ruth et al. 2014). The effective reflectivity was retrieved with eq (1) using the known cross-section of $NO_2$ and the mixing ratio from the CLS $NO_X$ monitor (e.g. using the measurements on 5 Oct 2011 where $R_{eff}$ = 0.9978 at 352 nm and $R_{eff}$ = 0.9986 at 386 nm, see supplementary Figure S5). The lower limit of the absolute uncertainty (~11%) is based on the accuracy of the $NO_2$ cross-section (8%) and that of the $NO_X$ measurement (7%).

The initial calibration measurement of $R_{eff}$ was also used to determine the optical loss, $L_{LLO}(\lambda)$, of an anti-reflection coated optic (referred to as "low loss optic", LLO), which was in turn used on a daily basis (in the morning) to determine the reflectivity in the clean chamber instead of using $NO_2$ as calibration gas (see also supplementary material and Figures S4 and S6) (Varma et al. 2009, Ruth et al. 2014):

$$R_{eff}(\lambda) = 1 - \left( \frac{I_{LLO}}{I_0 - I_{LLO}} L_{LLO}(\lambda) \right) \qquad (2)$$

In the transmitter unit the LLO ($R_{LLO}$ (375 nm) < 0.001, diameter 40 mm, parallelism 30") was moved into and out of the cavity parallel to the mirror by means of an accurate translational stage with high reproducibility (Ruth and Lynch 2008). Care was taken in designing the air-tight LLO compartment (Figure S1) where the optic was "parked" when not needed. The LLO was furthermore flushed with clean dry air while in the cavity to avoid (minimize) potential changes of its optical loss. When the LLO is used over the course of a simulation experiment, i.e. in a chamber with arbitrary gas mixture, the LLO measurements yield effective reflectivities that comprise extinction losses in the chamber at this particular time. Although only the $R_{eff}$ measured each morning in the clean dry and dark chamber was used to retrieve the concentrations of the target species (eq (1)) for the entire day, repeated introductions of the LLO into the cavity over the course of the day was used for checking whether misalignments or potential drifts of the cavity had occurred.

### 4.1.3 Data Evaluation

In order to judge the quality of the data evaluation an example of an IBBCEAS extinction spectrum and the corresponding fit of eq (1) to the measured data is shown in Figure 1 (uppermost panel). The data for this example were chosen because on 11 July 2011 HONO, $NO_2$ and MACR were simultaneously present in the chamber. The different panels in Figure 1 show the individual absorption contributions of the three species to the measured spectrum together with the polynomial background

determined in the fit analysis. The lowermost panel shows the fit residuals, $\Delta\epsilon$, illustrating the appropriate use of the reference absorption spectra of the three target species in this measurement.

### 4.1.4 Detection Limits

For individually measured spectra a minimum extinction coefficient of $9.3 \times 10^{-10}$ cm$^{-1}$ was determined for an acquisition time of 1 min (assuming a 1:1 signal-to-noise ratio), which corresponds to $\epsilon_{min} = 7.2 \times 10^{-9}$ cm$^{-1}$ Hz$^{-1/2}$ (1$\sigma$). For a series of measurements (see supplementary Figure S7) this translates into a measured 3$\sigma$ limit of detection (LOD) of the IBBCEAS instrument for a 1 min acquisition time of ~39 pptv for HONO, ~114 pptv for NO$_2$, and ca. 510 pptv for MACR in the 352-386 nm wavelength range.

In comparison to the IBBCEAS instrument, the LOPAP instrument at SAPHIR featured a 3$\sigma$ LOD of 10 pptv of HONO in 1 min. The LOD for NO$_2$ by the standardised chemiluminescence technique at SAPHIR also compares favourably with a 3$\sigma$ LOD of 13.4 pptv of NO$_2$ for a 1 min acquisition time with an overall accuracy of 7% (Rohrer and Brüning 1992, Fuchs et al. 2010) (the CLS detection limit for NO is $\approx$6.7 pptv). Mixing ratios of volatile organic compounds such as methacrolein can be only compared to data acquired with PTRMS, whose LOD for MACR is 8.5 pptv in 1 min (accuracy 8%).

An overview of (2$\sigma$) detection limits for HONO of previously published IBBCEAS instruments was recently given by Jordan and Osthoff 2020. The 2$\sigma$ detection limit of 26 pptv for HONO in 1 min presented here compares favourably to the IBBCEAS works by Gherman et al. 2008, Hoch et al. 2012, Wu et al. 2012, Donaldson et al. 2014, Scharko et al. 2014, Min et al. 2016, Yi et al. 2016, Nakashima and Sadanaga 2017, Duan et al. 2018, Jordan and Osthoff 2020, and Tang et al. 2020 - see Table 1 in the publication by Jordan and Osthoff, Atmos. Meas. Tech., 13, 273-285 (doi: 10.5194/amt-13-273-2020), and also Tang et al. 2020. A link to this table is given in the supplementary material together with a summary of experimental parameters specifying the current IBBCEAS setup. It should be noted, however, that some of the quoted detection limits were established under field conditions, which bears challenges that are different from experiments with a large scale simulation chamber. The low detection limits of the current setup are a result of the substantial cavity length, despite the fact that the effective cavity mirror reflectivity in this work was lower than in all other instruments reported.

### 4.2 LOPAP Instrument

In a previous DOAS-LOPAP intercomparison (Kleffmann et al. 2006) the addition of ozone triggered issues concerning the use of proper reference data in the DOAS evaluation procedure at ppbv levels of HONO. This sort of difficulty was not observed with IBBCEAS at sub-ppbv mixing ratios of HONO.

However, when switching from illuminated to dark conditions the LOPAP instrument regularly showed an increase of HONO concentrations, while the IBBCEAS detector does not seem to follow this trend as strongly and reproducibly; see Figure 2 (11 July 2011) or Figure S10, for example. The systematically increasing HONO mixing ratios measured by the LOPAP instrument upon closing the roof and generating dark conditions was also observed on a few other days during the summer campaign, e.g.

on 10 and 15 June (see supplementary Figure S10). The reason for this behavior is not fully understood yet. An effect of the presence of organic reaction products from the oxidation of mathacrolein that cause an interference with the LOPAP technique upon stopping photolytic processes can be ruled out, because the "effect" was also observed in experiments with a clean humidified chamber solely filled with high purity synthetic (zero) air. A possible interference of the LOPAP instrument towards $N_2O_5$ and/or $NO_3$, which are both formed in the dark chamber in the presence of $NO_2$ together with an excess of ozone, appears likely. However, laboratory investigations of Kleffmann et al. (2002, 2006, 2008) found no evidence for a cross sensitivity for $NO_3$ or $N_2O_5$ for a two-channel LOPAP instrument. This finding is supported by the IBBCEAS measurement data in our experiments which often show the same trend to elevated HONO concentrations in the dark after the roof was closed.

Another possible explanation may be based on the changing homogeneous and heterogeneous production rates of HONO versus its destruction rates under bright and dark conditions. For daytime conditions the steady state HONO mixing ratio is mainly determined by photolysis of a yet unknown nitrogen containing precursor adsorbed at the Teflon walls or dissolved in the surface water layer on the Teflon film versus the photolytic destruction of HONO into OH and NO. In the dark (closed) humidified chamber HONO can still be produced, but predominantly through heterogeneous reactions with formation rates being at least a factor of ~10 smaller than under illuminated conditions (i.e. with the roof open). Even though closing the roof stops efficient HONO production it also eliminates the photolytical loss entirely, while HONO heterogeneous production within the aqueous phase on the chamber walls is still active (albeit its slowly decreasing efficiency). This can lead to changing the equilibrium such that the subsequent outgassing of HONO from the aqueous phase on the Teflon film can temporarily dominate the HONO production into the gas phase until the initially HONO-saturated aqueous phase also gets depleted leading to the eventual decrease in HONO mixing ratios (Karl, 2004). The inlet of the LOPAP instrument is much closer to the chamber wall (ca. 30 cm) in comparison to the region probed by IBBCEAS, which averages across the whole length of the chamber near its center. Thus the effect is more likely to be apparent in the LOPAP data than in the IBBCEAS data. This tentative explanation of the observed trend upon closing the roof of the chamber warrants further investigation.

**4.3 CLS and PTRMS Instruments**

The CLS and PTRMS data were used here as reference guideline for the measurements and not for scrutinizing the performance of the respective instruments. Nevertheless, some discrepancies between IBBCEAS, CLS and PTRMS were noted and are outlined here.

**_CLS:_** Mixing ratios of $NO_2$ measured by CLS and IBBCEAS frequently appeared to differ somewhat during the humidification of the dark chamber. The CLS instrument appears to systematically record higher $NO_2$ mixing ratios than IBBCEAS upon humidification of an initially clean chamber without the obvious presence of $NO_2$. Examples of the observed behavior are shown in Figure 5 (see also Figure S10).

**Figure 5**

There are two possible scenarios that may explain these observations: (a) It could be an artefact in the chemiluminescence device, which may be due to a surface effect of the blue light converter (BLC). Switching from a dry to a humidified chamber leads to a severe change in the surface conditions of the BLC, which releases either NO or a substance that mimics an NO signal for some time until a new equilibrium has been reached. This type of behavior has occurred before in measurement campaigns at SAPHIR, where small temporary discrepancies between model calculations and measurement were observed. The nature of the potentially released substance is unclear. HONO would be more likely to stick to the walls rather than to be released unless heterogenous reactions are at play leading to the formation of NO. The effect is transient and vanishes after a short time. The fact that rapidly changing water vapor concentrations can affect the instrumental background of CLS detectors and lead to a non-trivial memory effect that cannot be easily corrected retrospectively has also been reported recently in the literature (Nussbaumer et al. 2021). (b) A small contamination of the milli-Q water and differences in the sample quality used for humidification cannot be fully ruled out. Those contaminations have also been observed before. The increase in the $NO_2$ signal upon humidification is supported by a significant increase in the NO signal (~120 pptv) observed by CLS. However, on 10 June 2011 this increase seems to be over after ca. 15 min while the humidification is still ongoing. The reason why the IBBCEAS instrument does not detect the increase in $NO_2$ if it is due to contamination is also unclear.

**_PTRMS:_** After introduction of MACR to the chamber on 11 July, initially higher mixing ratios were recorded by PTRMS rather than by IBBCEAS (between 11:50 hrs and ca. 15:30 hrs). The mixing ratios measured with the two instrument gradually approach very similar values over the said time period until they agree rather well when the roof of the chamber was closed. The reason for this behavior is not clear since no obvious interference in this period was present in the chamber.

## 4.4 Correlation of Data Obtained by Different Instrumental Methods

For the combined data sets an overall good agreement for both trend and absolute mixing ratios was observed between IBBCEAS and the established instruments at SAPHIR as illustrated in the correlation plots in Figure 6 and Table 1 (see also Figures 2(b), 3 and 4). The best agreement is observed on 5 Oct 2011 for both HONO and $NO_2$, which may be attributed to the fact that the least amount of interfering species were present in the chamber at that time. Correlation coefficients r range from 0.930 to 0.994 for HONO and between 0.937 and 0.992 for $NO_2$ (see also discussion at the end of section 4.5). For the single measurement of MACR a correlation coefficient of r = 0.981 is found between data from IBBCEAS and PTRMS.

**Figure 6**

## 4.5 Overview of other measurement campaigns concerning HONO detection

A number of HONO detection campaigns, in which different experimental approaches were used side-by-side, have been published in the literature (Kleffmann et al. 2006, Rodenas et al. 2013, Pinto et al. 2014, Wu et al. 2016, Crilley et al. 2019, Yi et al. 2021). Several of these campaigns include IBBCEAS and LOPAP experimental approaches and describe instrument

performances in the context of the measurement objectives for a given atmospheric (chamber of field) environment. An overview of the different campaign characteristics is summarized in Table 2 for information. Since the current study focussed on the development of a custom-designed long cavity open-path IBBCEAS setup for the SAPHIR chamber, the motivation and objectives of our work differ substantially from those of the majority of the published campaigns. Thus a detailed comparison of the observations is problematic, since the conditions and objectives are not compatible. Four of the studies listed in Table 2 are field campaigns with extractive (closed path) or no IBBCEAS instruments taking part. While the challenges in field work are substantial, the interfering species cannot be controlled. Pinto et al. 2014 pointed out that their instrument intercomparison in only valid and relevant for the specific field site in Texas where the study took place. Among the chamber studies only the latest one (Yi et al. 2021) uses an open-path IBBCEAS instrument. It is noteworthy that in contrast to the other chamber studies on HONO detection the measurements at SAPHIR were performed at HONO mixing ratios that are low and typical for urban environments, while the chamber measurements campaigns by Kleffmann et al. 2006, Rodenas et al. 2013, and Yi et al. 2021 used substantially higher mixing ratios (typically ~ ppbv range are reported). Since a multi-pass White cell type DOAS system was used in Kleffmann et al. 2006 and no data from open path IBBCEAS were published in the FIONA campaign, we will limit a brief comparison of our work with outcomes of the work by Yi et al. 2021:

In contrast to the measurements at SAPHIR (which is a 280 $m^3$ FEP teflon chamber) the measurements by Yi et al. 2021 were performed in a stainless steel chamber (CESAM, Créteil, France) with a volume of ca. 4.2 $m^3$. Therefore the open-path cavity length used at CESAM is ca. 10 times shorter than that used at SAPHIR. Owing to the longer cavity length at SAPHIR a consequently lower reflectivity could be used ($R$~0.9978-0.9986 (see Figure S5) leading to an effective pathlength of ~6.0 km in comparison to $R$~0.9992-0.9995 in Yi et al. 2021 leading to an effective pathlength of ~3.3 km. Yi et al. 2021 used a thermalized 300 mW UV-LED (at 365 nm) which covered the range 351-378 nm. However, the detection limits were evaluated for a fit range of only 362-372 nm, since the lower intensity "wings" of the LED spectrum contribute to more noise, which impinges negatively on the fit reliability. While the optical powers of the UV-LED and our arc lamp are largely comparable, depending on the light collecting optics (see specs in the supplementary information), the temporal stability of the LED and hence the constancy of $I_0(\lambda)$ can be expected be better than that of a hot-spot arc lamp. For cavity-enhanced absorption measurements in chamber studies this is a crucial criterion, since $I_0$ can only be established in the clean chamber and not easily during the experiments. Some unexpected variations in the HONO measurements in the current study are most likely attributed to the "hot spot" arc stability (e.g. see Figure 2(a) at 9:00 hrs). However, the limited spectral width of a single LED impinges on the achievable detection limits, which were ca. ~160 pptv for HONO and ca. ~340 pptv for $NO_2$ in Yi et al. 2021. These limits were amended for an integration time of 1 min and a 2σ standard deviation to make them comparable with our limits of 26 pptv for HONO and 76 pptv for $NO_2$. We attribute the smaller detection limits in the present study to two factors: (i) A larger spectral fit range, 352-386 nm, was used for the data evaluation. This range was determined by optical filters and the usable mirror reflectivity range. (ii) The effective absorption pathlength at SAPHIR was ca. twice that used at CESAM owing to the different sizes of the chambers. A third factor (iii) may be related to the lower mirror reflectivity in the SAPHIR experiment which in principle enables more light to be collected at the exit of the cavity. This advantage is however partly

compensated by the losses that occur in imaging light through a 10 times longer cavity. Argument (iii) is also biased by the use of different spectrometer/CCD assemblies in both experiments which can be expected to have different quantum yields in the relevant spectral range. The experiments at CESAM were executed at a different chemical regime with mixing ratios of HONO and $NO_2$ in the range of tens of ppbv with up to 30 ppbv for HONO and up to 120 ppbv for $NO_2$. In comparison, all SAPHIR measurements were executed at low mixing ratios of up to 400 pptv for HONO and up to 4 ppbv for $NO_2$, which is more realistic in terms of ambient field concentrations. At these low mixing ratios and with potential instabilities in the excitation light source the correlation of data from IBBCEAS and LOPAP (r = 0.974) and IBBCEAS and CLS (r = 0.985) are overall very satisfactory (see Figure 6 and Table 1). Despite the fact that Yi et al. 2021 used an expectedly more stable light source, the reported correlations coefficients of r = 0.977 (IBBCEAS-LOPAP) and r = 0.943 (IBBCEAS-FTIR) for HONO detection, and r =0.993 (IBBCEAS-CLS) and r = 0.941 (IBBCEAS-FTIR) for $NO_2$ detection are comparable. The correlation coefficients in the field study by Wu et al. 2014 are lower in comparison; i.e. r = 0.837 (IBBCEAS-LOPAP) and r = 0.906 (IBBCEAS-CLS).

## 5 Conclusion

In 2011 an incoherent broadband open-path cavity-enhanced absorption spectroscopy (IBBCEAS) instrument was established at the SAPHIR chamber in Jülich and optimized for the detection of HONO and $NO_2$ in the near UV region (352-386 nm) using a bright hot-spot Xe-arc lamp and a UV-enhanced CCD detector. Based on an effective reflectivity of $0.9978 < R_{eff} < 0.9986$ and a 20 m open-path cavity a $2\sigma$ detection limit of 26 pptv for HONO and 76 pptv for $NO_2$ was achieved for an integration time of 1 min. Methacrolein was also detected at mixing ratios below 5 ppbv. These detection limits are lower in comparison to those reported in the recent literature. The IBBCEAS instrument's performance for HONO and $NO_2$ detection was compared to that of long-path absorption photometry (LOPAP) and chemiluminescent $NO_X$ detection, with R coefficients range from 0.930 to 0.994 for HONO, and between 0.937 and 0.992 for $NO_2$, respectively. At low concentrations however, and especially upon humidification of the chamber, a small temporary water interference on $NO_2$ mixing ratios were observed in CLS measurements which was not observed in the IBBCEAS data.

*Data Availability.* The datasets used in this study are available from the corresponding author Albert A. Ruth upon request (a.ruth@ucc.ie).

*Author Contributions.* AAR and HPD designed and constructed the IBBCEAS setup. SD and HPD implemented the setup at SAPHIR. SD performed the IBBCEAS experiments (characterization, calibration, measurement, and data analysis) with advice from AAR and HPD. RH carried out the LOPAP measurements and analysis. FR carried out the CLS measurement and

analysis. TB operated the SAPHIR chamber and contributed to the running of the campaign experiments. AAR wrote the manuscript with contributions from SD (layout), HPD and FR.

*Competing Interests.* The authors declare that they have no conflict of interest.

*Special issue statement.* This article is part of the special issue "Simulation chambers as tools in atmospheric research
(AMT/ACP/GMD inter-journal SI)". It is not associated with a conference.

*Acknowledgements.* We thank Andreas Wahner, Director of the Institute for Energy and Climate Research: Troposphere (IEK-8) at the Research Center in Jülich (Germany), and the team running the SAPHIR chamber, especially Ralf Tillmann, for providing the PTRMS data for this publication. The extraordinary craftsmanship of the mechanical workshop at Jülich in the context of this project is also gratefully recognized.

*Financial Support.* SD was funded through an INSPIRE post-doctoral fellowship by the Irish Research Council for Science Engineering and Technology which was co-funded by the Marie Curie actions, COFUND, under the Seventh European Framework Programme (FP7). This project has received funding for open access publications costs from the European Union's Horizon 2020 research and innovation programme through the EUROCHAMP-2020 Infrastructure Activity under grant agreement No 730997. This publication has in parts emanated from research conducted with the financial support of Science
Foundation Ireland under Grant number 21/FFP-A/8973.

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

**Table 1. Comparison of correlation plot data.** *N*: number of data points, *a* and *b*: regression coefficients, σ (*a*) and σ (*b*): bootstrap errors of the linear square fit $y = a + bx$ (where *y* represent IBBCEAS data [pptv] and *x* the data [pptv] by either LOPAP, CLS or PTRMS), r : Pearson linear correlation coefficients.

| date | *N* | *a* [pptv] | σ(*a*) [pptv] | *b* | σ(*b*) | r |
|---|---|---|---|---|---|---|
| HONO: IBBCEAS vs. LOPAP | | | | | | |
| 11 July | 553 | 11.1 | 2.5 | 0.915 | 0.013 | 0.9404 |
| 12 July | 360 | –0.3 | 1.3 | 0.883 | 0.017 | 0.9298 |
| 5 Oct | 299 | –16.2 | 1.1 | 0.992 | 0.007 | 0.9942 |
| 6 Oct | 394 | –26.1 | 2.9 | 1.020 | 0.013 | 0.9700 |
| **all 4 days** | **1668** | **–8.4** | **0.9** | **0.973** | **0.005** | **0.9741** |
| NO$_2$: IBBCEAS vs. CLS | | | | | | |
| 11 July | 708 | –157 | 11 | 1.085 | 0.012 | 0.9618 |
| 12 July | 378 | –56 | 5 | 1.090 | 0.019 | 0.9703 |
| 5 Oct | 531 | 15 | 7 | 1.050 | 0.011 | 0.9918 |
| 6 Oct | 634 | 3 | 2 | 1.385 | 0.019 | 0.9373 |
| **all 4 days** | **3011** | **–29** | **3** | **1.031** | **0.007** | **0.9849** |
| MACR: IBBCEAS vs. PTRMS | | | | | | |
| 11 July | 657 | 103 | 27 | 0.947 | 0.010 | 0.9810 |




**Table 2. Overview of other HONO detection campaigns in the literature in chronological order. The grey columns represent chamber studies, the white columns refer to field campaigns (Kleffmann et al. 2006 report measurements of both).**

| | Kleffmann et al. 2006 | Rodenas et al. 2013 | Pinto et al. 2014 | Wu et al. 2014 | Crilley et al. 2019 | Yi et al. 2021 | Present work |
|---|---|---|---|---|---|---|---|
| Time of data acquisition | 2001/2002/2004 | July 2009 | Apr 2009 | May 2012 | Nov 2016 | 2020 | Jul/Oct 2011 |
| Chamber / location | EUPHORE Valentia, Spain | EUPHORE Valentia, Spain | | | | CESAM Créteil, France | SAPHIR Jülich, Germany |
| Field site location | Milan, Italy[1] | | Houston, USA[2] | Hong Kong, China[3] | Beijing, China[4] | | |
| Formal | no | no | in parts | no | yes | no | no |
| HONO range[5] [ppbv] | Chamber ~0 ..6 Field ~0...3 | ~0…25 | ~0…1.5 | ~0…2 | ~0…9 | ~0…30 | ~0…0.4 |
| NO$_2$ range[5] [ppbv] | Chamber Field ~0...175 | | | ~0…60 | | ~0…120 | ~0…4 |
| **Techniques[6] for target species HONO and NO$_2$** | | | | | | | |
| **IBBCEAS** | n/a | HONO, NO$_2$ ×3 | n/a | HONO, NO$_2$ | HONO, NO$_2$ ×2 | HONO, NO$_2$ | HONO, NO$_2$, MACR |
| **LOPAP** | HONO | HONO ×6 | HONO | HONO | HONO ×2 | HONO | HONO |
| CLS | | | | NO$_2$ | | NO$_2$ | NO$_2$ |
| (LP)-DOAS | HONO, NO$_2$ | HONO, NO$_2$ | HONO | | | | |
| DOAS (White cell) | HONO, NO$_2$ | | | | | | |
| FTIR | | HONO, NO$_2$ | | | | HONO, NO$_2$ | |
| SC-AP | | | HONO | | | | |
| MC-IC | | | HONO | | | | |
| QC-TILDAS | | | HONO, NO$_2$ | | | | |
| IC-CIMS | | | HONO | | | | |
| TOF-CIMS | | | | | HONO | | |
| SIFT-MS | | | | | HONO | | |
| CIMS | | HONO | | | | | |
| PF-LIF | | HONO | | | | | |
| LC-MS | | HONO | | | | | |
| **IBBCEAS instrument overview** | | | | | | | |
| **IBBCEAS** | n/a | Extractive | n/a | Extractive[7] | Extractive[8] Extractive[9] | Open path | Open path |
| Spectral region [nm] | | 355-390 | | 352-376 | 359-387[8] 362-374[9] | 362-372 | 352-386 |
| 2σ LOD [pptv] HONO (60 s) | | No info | | | 120[8] ~50[9] | ~160 | 26 |
| 2σ LOD [pptv] NO$_2$ (60 s) | | No info | | | | ~340 | 76 |
| **Species that were reported/considered in campaign data apart from HONO and NO$_2$** | | | | | | | |
| | H$_2$CO, O$_3$, NO, n-butane, ethene, toluene | aerosols, nitrophenols | benzene,toluene,C2-C3 alkylbenzene, | NO, CO, O$_3$, SO$_2$ | NO | H$_2$O, H$_2$CO | MACR, NO, H$_2$O, CO, isoprene |

| | | | | | | | |
|---|---|---|---|---|---|---|---|
| | | nitrites, inorganic nitrates, $O_3$, aerosol, aromatics, peroxides, $SO_2$, small aldehydes | acetaldehyde, $H_2CO$, $O_3$, $HNO_3$, $NO_3$, PAN, $H_2O_2$ $CH_3OOH$, $H_2O_2$, phenol, styrene, iso-prene, mono-terpenes | | | | |
| Air type of field site | urban (background) | | urban | urban | urban | | |

[1] Bresso, northern outskirts of Milan 5 km north of the city centre
    [2] Moody Tower at University of Houston 4 km south-East of Houston city centre
    [3] Suburban town of Tung Chung in Southwestern Hong Kong, adjacent to Hong Kong International Airport
    [4] Chinese Academy of Sciences' Institute of Atmos. Physics (IAP) tower campus near the 4th ring road in northern Beijing
    [5] zero here refers to "below the detection limit"

[6] The types of techniques mentioned in the table is limited to HONO and $NO_2$ detection for comparison reasons.

**IBBCEAS:** Incoherent broadband cavity enhanced absorption spectroscopy

**LOPAP:** Long path absorption photometry

CLS: Chemiluminescence spectroscopy

PTRMS: Proton transfer reaction mass spectrometry

(LP)-DOAS: (Long path)-differential optical absorption spectroscopy

FTIR: Fourier transform infrared spectroscopy

SC-AP: Stripping coil-visible absorption photometry

MC-IC: Mist chamber/ion chromatography

QC-TILDAS: Quantum cascade-tunable infrared laser differential absorption spectroscopy

IC-CIMS: Ion drift-chemical ionization mass spectrometry

TOF-CIMS: Time-of-flight chemical ionization mass spectrometer

SIFT-MS: Selected ion flow tube mass spectrometer

PF-LIF: Photo fragmentation laser-induced fluorescence

LC-MS: Liquid chromatography-mass spectrometer

[7] Extractive (enclosed path) version of the open-path instrument in Wu et al. (2012)
    [8] Instrument from Duan et al. (2018)
    [9] Instrument reported in Kennedy et al. (2011), adapted for the detection of HONO and $NO_2$.

$\times n$ indicates the number of distinct instruments used for a particular measurement approach.



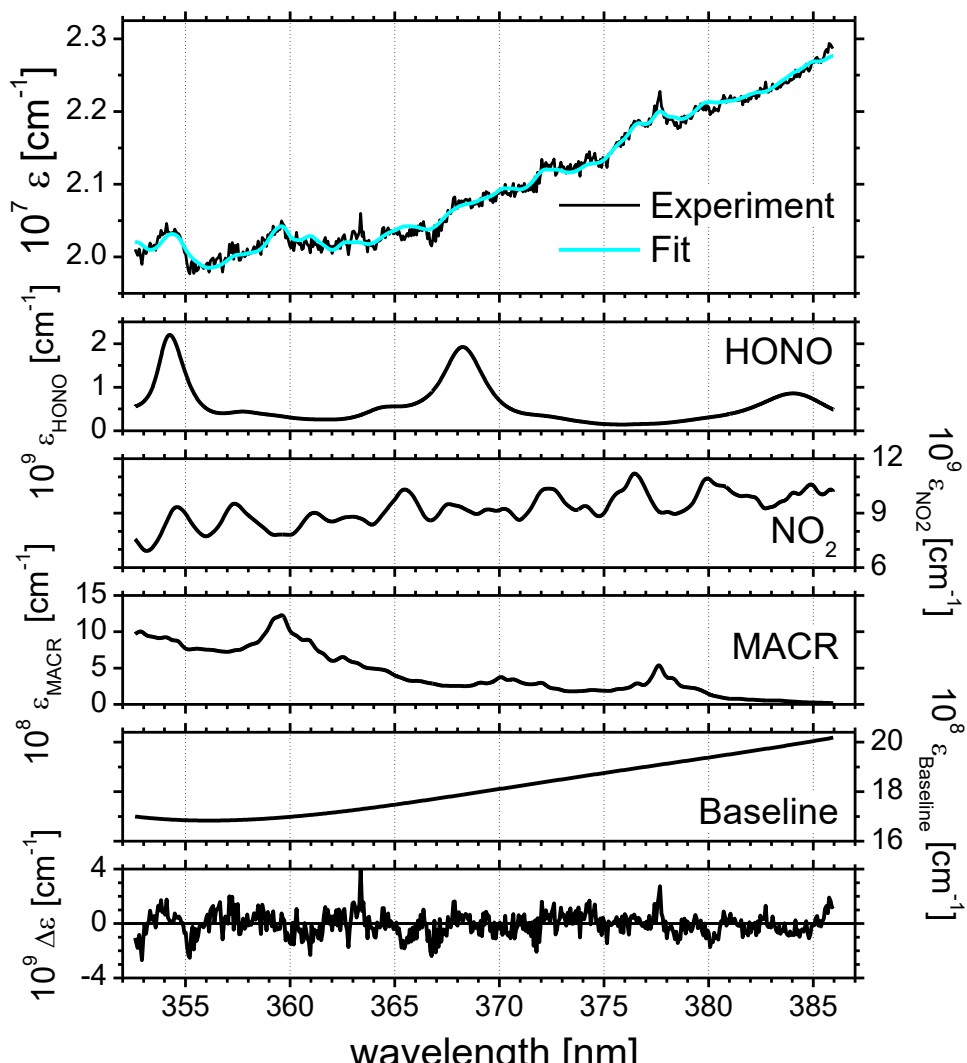


**Figure 1: Example of the extinction coefficient spectrum, $\varepsilon(\lambda)$, measured with IBBCEAS at SAPHIR in the wavelength range 352-386 nm with an integration time of 1 min, taken at 13:00 hrs on 11 July 2011. Uppermost panel: measured extinction (black), fit of eq (1) to the extinction (cyan). Lowermost panel: Absolute fit residuals $\Delta\varepsilon = \varepsilon_{fit} - \varepsilon_{exp}$. HONO, NO$_2$ and MACR mixing ratios were**
**retrieved as $n_{HONO} = 0.160$ ppbv, $n_{NO2} = 0.586$ ppbv and $n_{MACR} = 8.055$ ppbv, respectively. The corresponding contributions and the polynomial baseline are shown in the middle panels. The absolute wavelength scale was calibrated with a low-pressure Ne pen ray lamp (see Figure S9). No O$_2$-O$_2$ absorption at ~360 and ~380 nm is observed since all spectra, $I_0(\lambda)$ and $I(\lambda)$, were measured in synthetic air.**


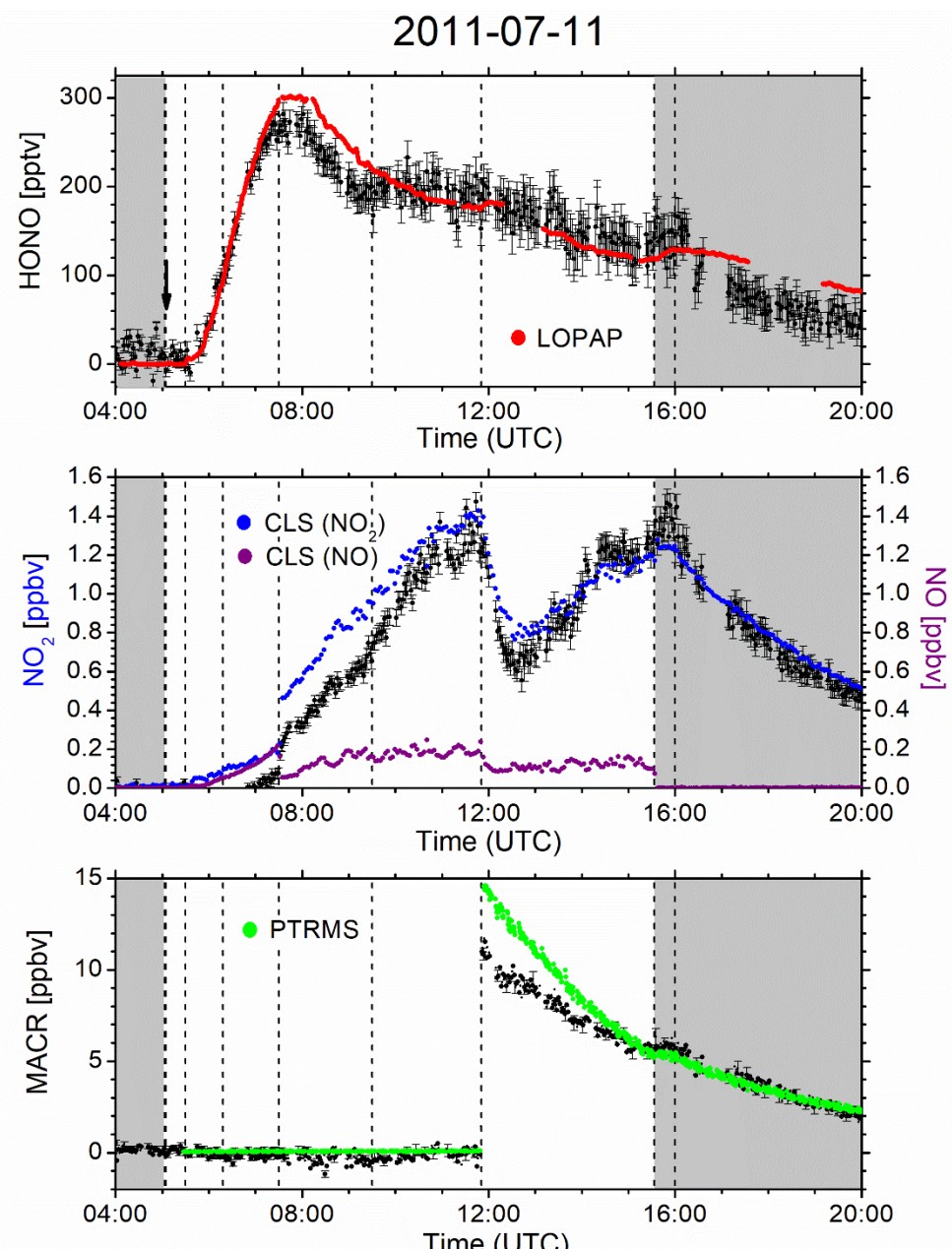

**Figure 2(a):** Time-dependent HONO, NO₂ and MACR mixing ratios measured by IBBCEAS (black trace) at SAPHIR on 11 July 2011. The NO mixing ratios (purple) measured by the CLS instrument are also shown. Dashed vertical lines indicate changes in the chamber conditions according to the experimental protocol: Overnight flushing of chamber with zero air stopped (5:03), roof opened (5:05), start of humidification (5:30), end of humidification (6:18), 40 ppbv $O_3$ (7:30), 750 ppbv CO (09:30), MACR (11:51), roof closing (15:33), flushing with zero air started (16:00). IBBCEAS data taken in the near-UV region of the spectrum (352-386 nm) are compared to LOPAP (red), CLS (blue) and PTRMS (green) data. The vertical black arrow indicates the time when $I_0$ was measured (generally for 10 min).


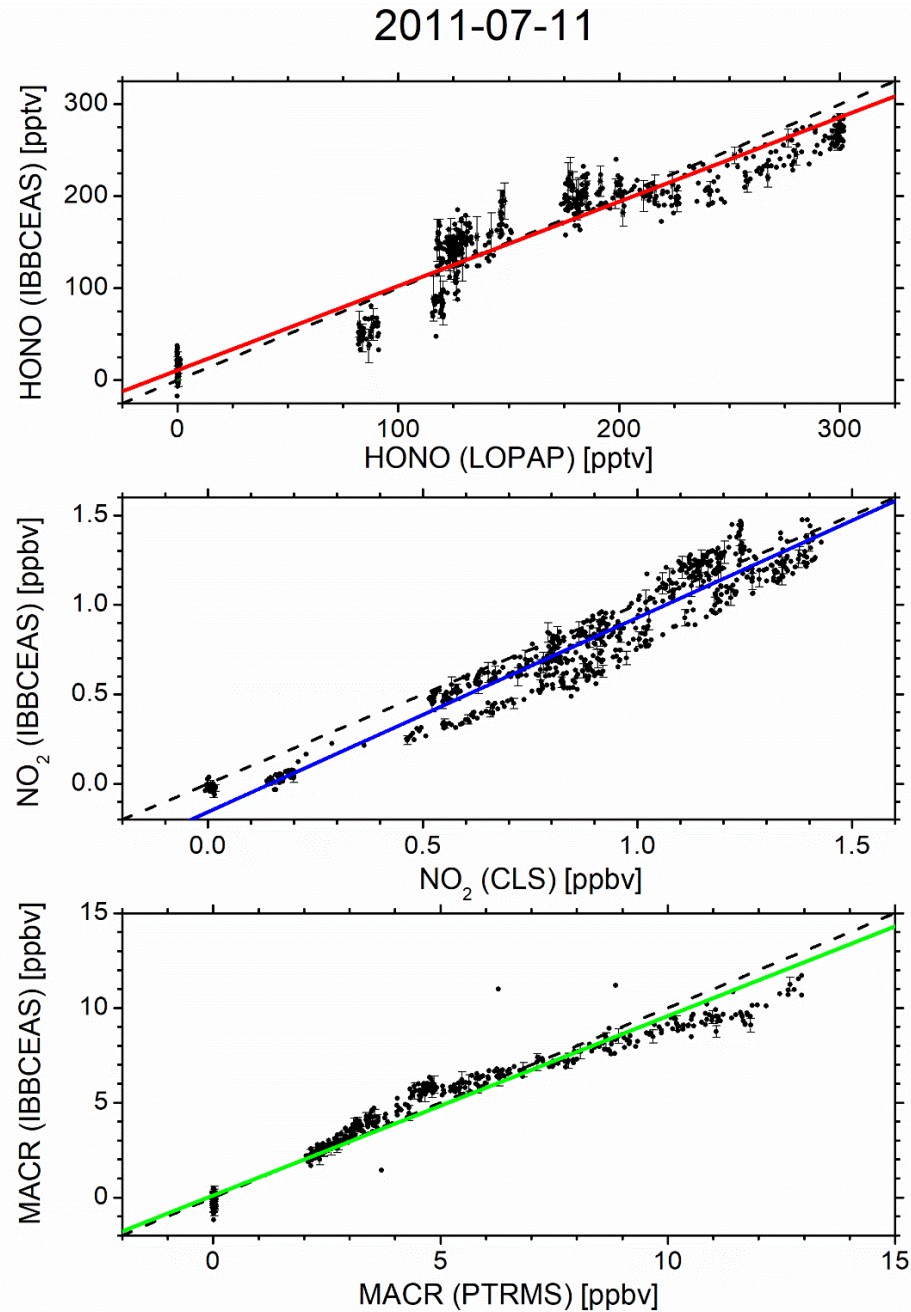

**Figure 2(b): Correlation plots of two instruments being compared against IBBCEAS. The dashed line represents the identity, the colored solid lines are linear regressions to the data. Results are listed in Table 1.**

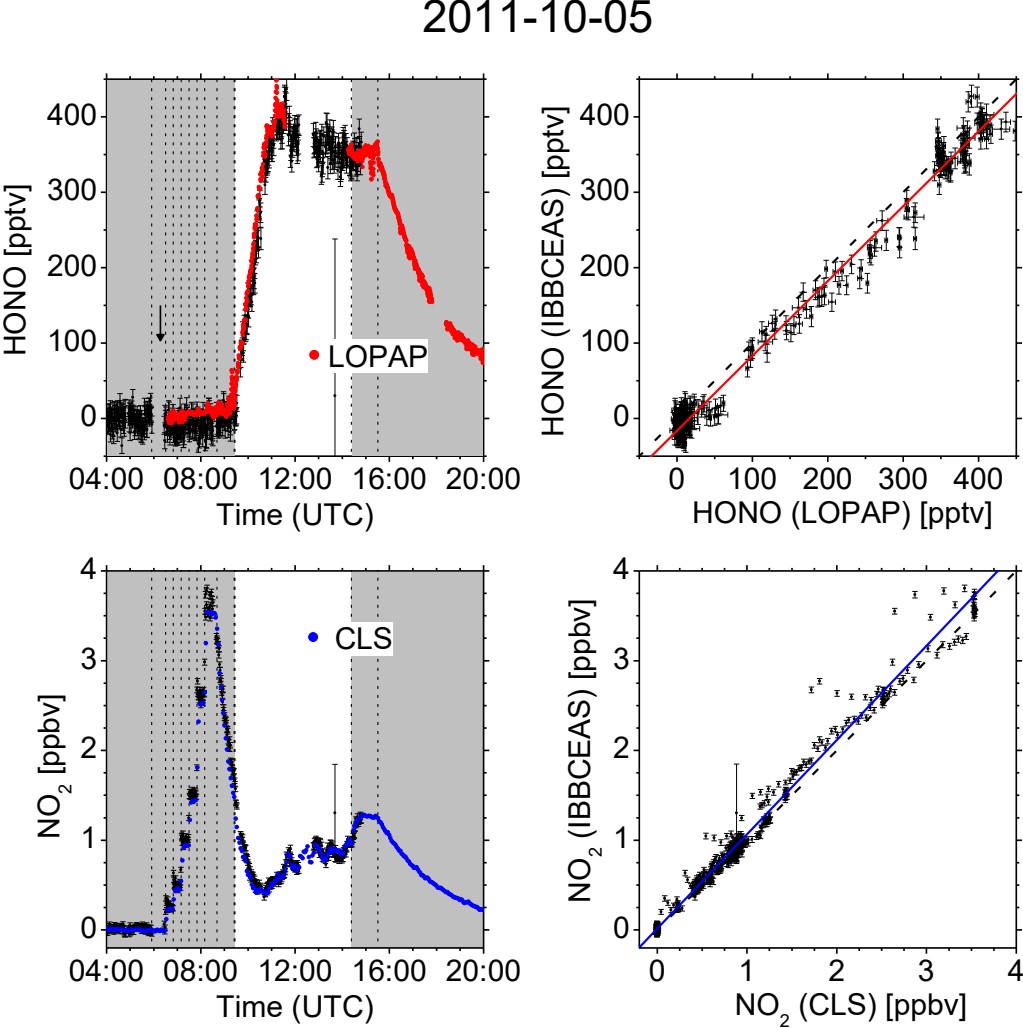


**Figure 3:** *Left panels:* **Time-dependent HONO and NO₂ mixing ratios measured at SAPHIR on 5 October 2011. Dashed vertical lines indicate changes in the chamber conditions according to the experimental protocol: Overnight flushing stopped (5:55), 250 pptv NO₂ (6:30 and 6:50), 500 pptv NO₂ (7:10 and 7:30), 1 ppbv NO₂ (7:50 and 8:10), start humidification (8:41), stop humidification (9:25),**
**roof opening (9:27), roof closing (14:23), flushing started (15:31). IBBCEAS data taken in the near-UV region of the spectrum (352-386 nm) are compared to LOPAP (red) and CLS (blue). The vertical black arrow indicates the time when $I_0$ was measured (generally for 10 min).** *Right panels:* **Correlation plots of two instruments being compared against IBBCEAS. The dashed line represents the identity, the colored solid lines are linear regressions to the data. Results are listed in Table 1.**

## 2011-10-06

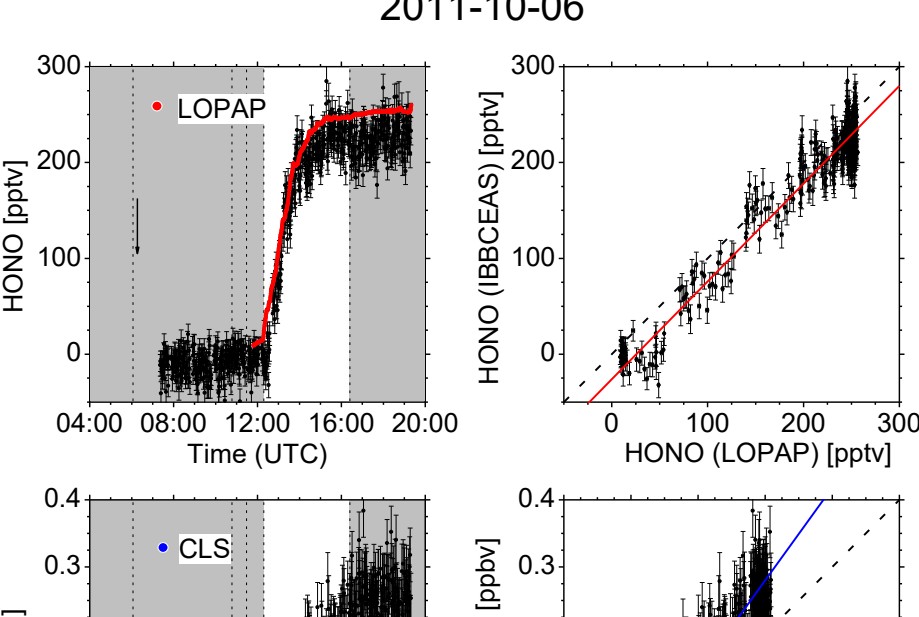


**Figure 4:** *Left panels:* **Time-dependent HONO and NO₂ mixing ratios measured at SAPHIR on 6 October 2011. Dashed vertical lines indicate changes in the chamber conditions according to the experimental protocol: Overnight flushing stopped (6:04), start humidification (10:47), stop humidification (11:29), roof opening (12:18), roof closing (16:24). IBBCEAS data taken in the near-UV region of the spectrum (352-386 nm) are compared to LOPAP (red) and CLS (blue) data. The vertical black arrow indicates the**
**time when $I_0$ was measured (generally for 10 min).** *Right panels:* **Correlation plots of two instruments being compared against IBBCEAS. The dashed line represents the identity, the colored solid lines are linear regressions to the data. Results are listed in Table 1.**


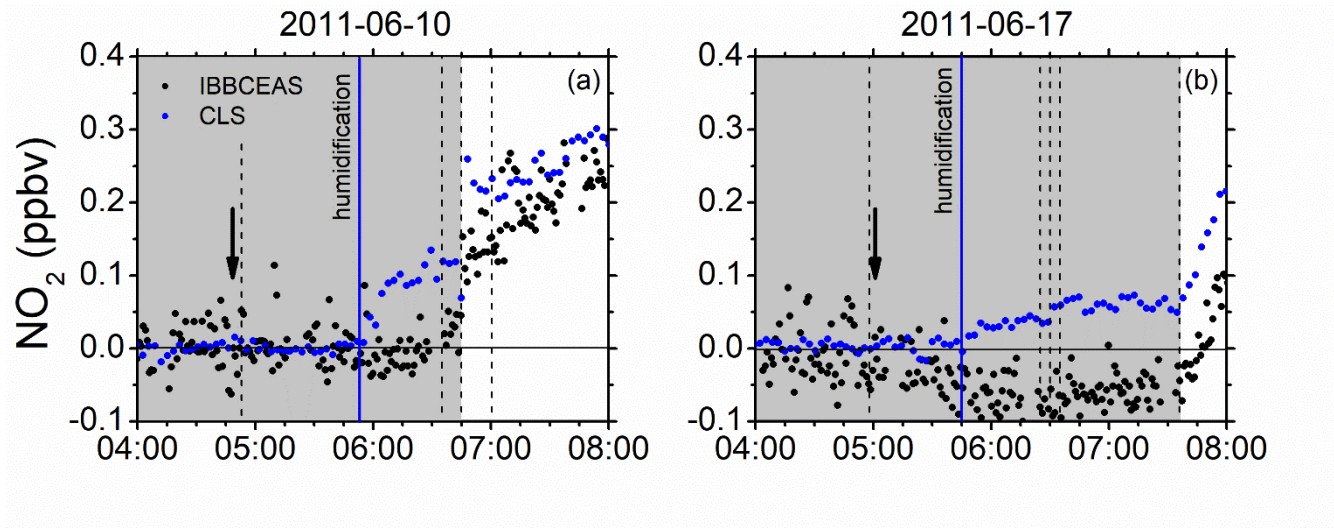

**Figure 5: Examples of the small increase of NO₂ mixing ratios upon humidification in the *dark* chamber measured with CLS (blue dots) in comparison to those measured with IBBCEAS (black dots): (a) 10 June 2011: CO₂ addition (4:53), start of humidification (5:52 - vertical solid blue line), ozone addition (6:36), roof opened (6:46), isoprene addition (7:01). (b) 17 June 2011: Flushing of chamber stopped (04:58), start of humidification (05:45), end of humidification (06:25), O₃ addition (06:30) in dark chamber, CO addition (06:35), roof opened (07:36). Vertical solid arrows indicate the time of the zeroing measurement ($I_0$).**

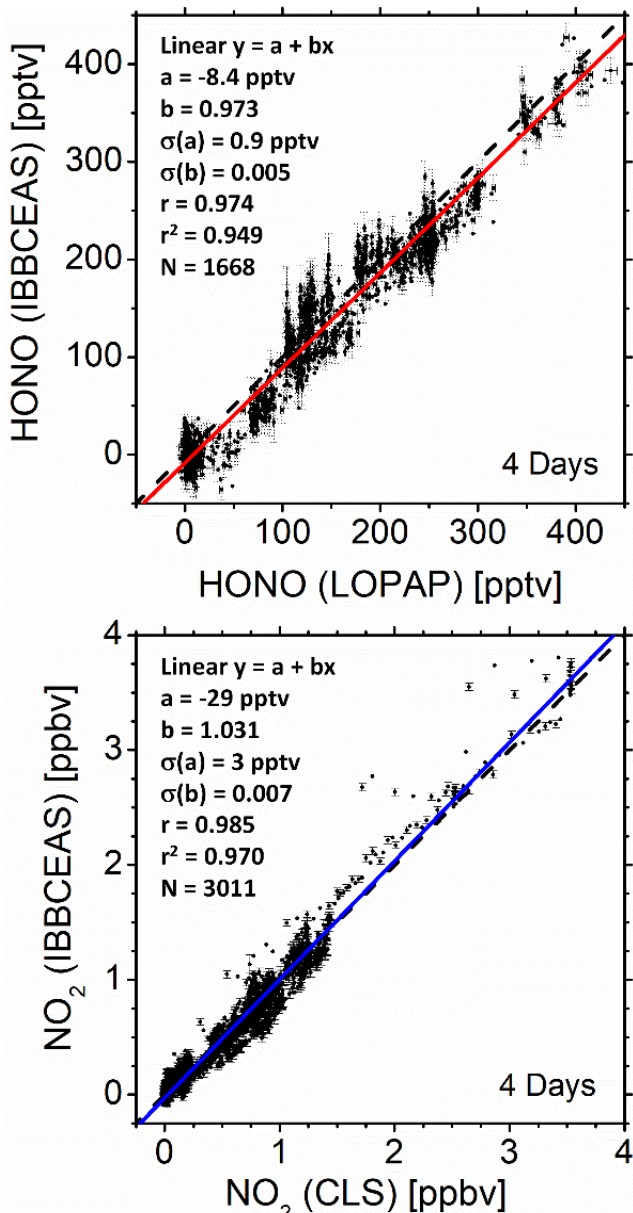

**Figure 6: Correlation plots of two instruments of the entire data set from the intercomparison. The dashed line represents the identity, the coloured solid lines are linear regressions to the data (see also Table 1).**