# Peer review of "Detection of nitrous acid in the atmospheric simulation chamber SAPHIR using open-path incoherent broadband cavity-enhanced absorption spectroscopy and extractive long-path absorption photometry"

_Atmospheric Measurement Techniques, 2021_

## Author Comment (AC1)

**RC1:  Reply to Anonymous Reviewer #2   (12 Dec 2021)**

Dixneuf et al. present new measurements of nitrous acid, nitrogen dioxide and methacrolein at the Jülich SAPHIR chamber by open-path incoherent broadband cavity-enhanced absorption spectroscopy (IBBCEAS), chemiluminescence spectrometry, long-path absorption photometry (LOPAP) and proton-transfer mass spectrometry (PTR-MS). The measurement of these analytes by these techniques have been described in the literature (i.e., are not novel) but there is nevertheless value in the comparisons and technical details presented.

Overall, the manuscript should be acceptable for publication once the authors have considered and addressed my comments below.

**Reply:** We would like to thank the reviewer for this thorough assessment of the manuscript and appreciate the observations and interesting comments made. In the large majority of cases we agree with the suggestions made, and tried to implement and improve the manuscript as good as possible and reasonable.

General comments:

**RC1:**
(1) The main focus of this paper is on the HONO intercomparison. There have been several such papers in the literature (e.g., Kleffmann et al., Atmos. Meas. Tech., 14, 5701-5715, 10.5194/amt-14-5701-2021, Rodenas et al., 2013, doi: 10.1007/978-94-007-5034-0_4 2021; Pinto et al., JGR 119, 5583, 2014, doi: 10.1002/2013JD020287; Wu et al., Atmospheric Environment 95, 544-551, 2014, doi: 10.1016/j.atmosenv.2014.07.016; Crilley et al., Atmos. Meas. Tech., 12, 6449-6463, 10.5194/amt-12-6449-2019, 2019; Yi et al. Atmos. Meas. Tech., 14, 5701-5715, 10.5194/amt-14-5701-2021, 2021). What was learned in this work that confirmed or was new or different from these prior intercomparisons? It seems an entire paragraph of discussion should be added (at least).

**Reply:** We really appreciate this comment, because it is important for the understanding of the nature of this article. We learned from this comment that based on the current title of our manuscript the motivation for our work can be easily misunderstood. As a consequence of this observation by the reviewer, we have come to the conclusion that the title of the manuscript should be rephrased to avoid any ambiguity concerning the intention of our work. We therefore decided to change the title to:

"*Detection of nitrous acid in the atmospheric simulation chamber SAPHIR using open-path incoherent broadband cavity-enhanced absorption spectroscopy and extractive long-path absorption photometry*".

With a validation instrument (LOPAP) at hand at SAPHIR, our experimental motivation at the time of the experiments were:

(1) To set up a long cavity open path IBBCEAS instrument for the SAPHIR chamber for the detection of HONO (proof of principle; with $NO_2$ and MACR being side aspects).

(2) To test the performance of the instrument in view of its robustness concerning its mechanical components (e.g. vibrations or temperature drifts), and in terms of its stability with regard to the light source and the coupling of light into the long cavity.

(3) To establish the detection limits that can be achieved with this long cavity and medium mirror reflectivity.

(4) To see how the data compare with the LOPAP results. Since the spectroscopic data evaluation in IBBCEAS also yield mixing ratios of $NO_2$ and MACR the validation of the corresponding data with results from CLS and PTRMS was obvious.

Thus reporting on the features of the IBBCEAS instrument and comparing the performance to the Jülich LOPAP is at the core of this publication. It should be noted that the reported measurements were performed (however not published) in 2011 following the FIONA campaign in 2009 (Rodenas et al. 2013), in which no open path IBBCEAS data were published. Apart from Gherman et al. 2008, where no validation instrument for HONO detection was available, the only directly comparable measurements are indeed from measurements in 2020 (Yi et al. 2021). Other campaigns use extractive IBBCEAS systems or no CEAS instruments at all, and/or instruments were part of field campaigns where interfering species cannot be controlled. The IBBCEAS instrument reported here is not adaptable to field measurements; it was designed to give the SAPHIR facility the capability to detect HONO with high sensitivity – this fact was known at the beginning of this project.

Having said that, we acknowledge that time has moved on and that the paper was indeed lacking the recognition of other campaigns on HONO detection, be they comparable or not. Therefore we included Table 2 to the manuscript, which gives an overview of the nature and characteristics of other campaigns on HONO detection that have been published in the past. A detailed comparison is however really only meaningful with Yi et al. 2021, as the corresponding research targets are most similar to our approach. Section 4.5 has been added to the manuscript to that avail.

In addition, a more explicit comparison of the IBBCEAS specifications with those listed in Jordan and Osthoff, 2020 (see Table 1 in Jordan and Osthoff) was also included at the end of the supplementary material.

**RC1:**
(2) The quantification of methacrolein by DOAS and validation vs. PTR-MS is novel (it was news to me, at least) and should be presented in more detail and discussed.
**Reply:** We expanded Figure 1 (now Figure 2) to put an emphasis on the measurements concerning methacrolein (MACR). Figure 2 has now two parts (A) and (B). A paragraph on the significance of MACR was included in the introduction and some more information was included at the end of Section 3.1:
"The ability of the IBBCEAS instrument to measure MACR selectively and with high sensitivity is a useful addition for the evaluation of photochemical experiments of the radical driven oxidation of isoprene, since the detection of the jointly formed reaction products MACR and MVK (methyl vinyl ketone) by mass spectrometric techniques using $H_3O^+$ as reactive ion can only determine the sum of MACR and MVK in the gas phase."

**RC1:**
(3) There has been some literature that could aid in the interpretation of some results. For example, Rohrer et al. (ACP 5, 2189, 2005, www.atmos-chem-phys.org/acp/5/2189/) in their Figures 1 and 2 (top left) show LOPAP HONO data larger than calculated HONO concentrations that continued (and in one case drastically increased) "in the dark" which should be mentioned in section 4.2 of this paper. Given this history, perhaps a more critical discussion of LOPAP performance is warranted.
**Reply:** The observation concerning the paper by Rohrer et al. 2005 has been addressed in the next reply below. A more critical discussion of LOPAP has been addressed in the context of a possible interference towards towards $N_2O_5$ and/or $NO_3$ under dark conditions (see comment on potential interferences below).

**RC1:**

(4) Rohrer et al. (ACP 5, 2189, 2005, www.atmos-chem-phys.org/acp/5/2189/) box-modeled the evolution of HONO mixing ratios. Why was this not done here? This would have been useful for the interpretation of LOPAP and IBBCEAS data, especially when diverge.

**Reply:** Rohrer et al. 2005 already published the observation of the occurrence of HONO and $NO_2$ during the humidification process. In that publication, a simple box-model was used to calculate the time evolution of HONO and $NO_2$ during humidification. This was just an empirical approach by introducing an assumed constant production processes for these species into the model mechanism. The assumed process started at the beginning and ended at the termination of the humidification period. There was no detailed physical or chemical theory behind that assumed production process. Its purpose was to adjust the calculated HONO and $NO_2$ concentrations at the end of the humidification period so that the following production of NO and OH by photolysis of HONO and the resulting $O_3$ generation could be explained. Therefore the parameters of that production process were set day by day to match the observations at the end of the humidification period. In the current paper it would not help to determine which instrument is more realistic. See Rohrer et al. 2005, Figure 1:"In the time interval between 07:00 and 08:30 UTC, small amounts of HONO and $NO_2$ were flushed into the chamber during the humidification process."

**RC1:**

(5) Re IBBCEAS data reduction - the background (R measurement) was taken at the beginning of each day. Have the authors considered averaging between consecutive measurements to reduce instrument drift. Some data (e.g., June 10, Figure S8) suggest that drift could be a substantial factor!

**Reply:** The daily measurement procedure comprised the measurement of $I_0(\lambda)$ and $R(\lambda)$ in the clean and dry chamber in the morning. On most days $R(\lambda)$ was measured using a calibrated low loss optic (LLO) rather than using $NO_2$. During the day with sample mixtures in the chamber, measurements of the reflectivity were performed regularly to check for potential drifts using the low loss optic. Due to the mirror purge the reflectivity of the mirrors did not degrade significantly during the entire measurement campaign. It turned out that the best results in comparison to the LOPAP and CLS instruments were obtained by using the morning parameters of $I_0(\lambda)$ and $R(\lambda)$.The drift during the day was random and averaging over reflectivity measurements during the day yielded no improvements of the results (see figure below showing $R$ measurements based on the LLO at different time 1-10; also compare with Figures. S5 and S6 in the supplementary material).

[Figure]

Averaging over several days was not attempted since re-optimization of the optical alignment was part of the daily measurement procedure. Moreover, the frequency of $R$ measurements would have been lower and the average would have to account for even longer time periods. The most critical aspects in this work is the stabilization of the "hot spot" arc by mean of a quadrant detector and piezo control unit. Since the reflectivity check using the LLO is dependent on $I_0$, the fluctuations displayed in the figure include the stability of the Xe arc. A second spectrometer (Avantes) in the transmitter unit, not shown in Figure S1, was additionally used to check for the drift/fluctuation of the lamp intensity. It was omitted from the description because any corrections of the intensity using this second spectrometer indeed were negligible.

Stray light measurements were also performed regularly (in the morning and during the day) to establish the true dark signal, especially before opening and closing the chamber roof. Its systematic influence turned out to be negligible.

Finally, the mechanical stability of the transmitter and receiver units proved to be very good, despite the challenge of a 21 m long cavity. The cavity alignment turned out to be surprisingly stable and the general mechanical layout was much improved in comparison to previous work (Varma et al. 2009).

Minor comments:

**RC1:**
Line 20, abstract: "methacrolein was detected at mixing ratios <5 ppbv" It would be more informative if the LOD for methacrolein was stated (could be combined with the previous sentence) rather than burying this information in Figure S7.
**Reply:** We included the estimated $2\sigma$ detection limit in the abstract. The following sentence:
"Methacrolein was also detected at mixing ratios below 5 ppbv."
was changed into,
"Methacrolein (MACR) was also detected at mixing ratios below 5 ppbv with an estimated $2\sigma$ detection limit of 340 pptv for the same integration time."
The information on this detection limit was not buried in Figure S7 but stated in section 4.1.4 on detection limits (originally in line 316/317).

**RC1:**
Line 20-22, abstract: Please state what the results of the comparisons were.
**Reply:** We added the following text to the abstract:
"For the combined data sets an overall good agreement for both trend and absolute mixing ratios was observed between IBBCEAS and these established instruments at SAPHIR. Correlation coefficients r for HONO range from 0.930 to 0.994 and for $NO_2$ from 0.937 to 0.992. For the single measurement of MACR r = 0.981 is found in comparison to proton transfer reaction – mass spectrometry (PTRMS)."

**RC1:**
Line 57. "sampling lines are known to cause unreliable results" is too strong a statement since inlets can be operated (e.g., with inert materials such as Teflon and fast flow rates, etc.) to suppress inlet wall artifacts. Consider rephrasing to "sampling lines can cause unreliable results"
**Reply:** Agreed and changed.

**RC1:**
Line 59. "E.g. validation" please correct grammar
**Reply:** We wrote "For example, validation…"

**RC1:**
Line 63. Just curious - is this long-path instrument still set up, or has it been dismantled?
**Reply:** The IBBCEAS setup has been dismantled in the meantime.

**RC1:**
Line 78. replace "detection" with quantification or measurement
**Reply:** Done.

**RC1:**
Line 84. The SAPHIR chamber has been described elsewhere - consider citing these papers (e.g., Fuchs et al.)
**Reply:** We included "(see e.g. Rohrer et al. 2005)" after the sentence.

**RC1:**
Line 95. Please define r and R (I assume they are radius of curvature and reflectivity) and use symbols different from the Pearson correlation coefficient (R, e.g., line 393).
**Reply:** We corrected the symbol used in lines 392 and 393, this ambiguity was well spotted. We think that otherwise the difference between the correlation coefficient and the reflectivity is pretty unambiguous, since a small letter r (non-italics) was used for the correlation coefficient and in a completely different context. We nevertheless spelled out the information in line 95 rather than using symbols.

**RC1:**
Please comment on why mirrors with a negative r were chosen since (I believe) concave mirrors are more commonly used.
**Reply:** Concave mirrors were indeed used. The minus sign comes from the common sign convention in optics. Distances measured in the direction of incident light are taken as positive, while distances measured in a direction opposite to the direction of the incident light are taken as negative. Opticians specify the radius of curvature with a negative sign. We included the word "concave" in the text to be unambiguous.

**RC1:**
Also, since mirror reflectivity was measured more accurately in section 4, consider calling out this section here (rather than on line 143).
**Reply:** We are merely specifying the components here and describing the setup, rather than the procedure/method how to establish the reflectivity. We think no further change is needed here.

**RC1:**
Line 134. The choice of $NO_2$ reference cross-sections (Merienne et al. 1995) differs from what was chosen by other IBBCEAS users in this wavelength region, with Burrows et al. (1998) and Voigt et al. (2002) being popular. Considering the possibility of HONO impurities in the $NO_2$ reference spectrum, which can lead to negative interference at low $HONO/NO_2$ ratios as in this work (Kleffmann et al., 2006), a justification to choose the cross-sections by Merienne et al. is warranted, along with a discussion of this potential interference.

**Reply:** Several reference spectra from the literature (Bogumil et al. 2003, Burrows et al. 1998, Harder et al. 1997, Mérienne et al. 1995, Vandaele et al. 2002, Voigt et al. 2002; see satellite.mpic.de/spectral_atlas/cross_sections/Nitrogen%20oxides/NO2.spc) were used to model the $NO_2$ absorption spectra in the relevant spectral range. The reference data yielding the smallest residuals in measurements with only $NO_2$ were chosen as reference cross-sections in measurements with gas mixtures – this turned out to be Mérienne et al. 1995. The data by Mérienne et al. 1995 were used by Kleffmann et al. 2006, who pointed out that reference spectra for absorption cross-sections of $NO_2$ may be biased due to the production of small amounts of HONO in the $NO_2$ reference samples. While this cautioning by Kleffmann et al. is in principle reasonable, in our retrieval the cross-sections by Mérienne et al. 1995 did not appear to deliver substantially different results than the reference spectra by authors cited above. The standard deviation of the least square discrepancies between the different cross-sections was ~ 5.5%, including Mérienne et al. 1995. This is below the uncertainty limit stated in Section 4.1.2. A short text to this avail was also added to the manuscript in section 2.1. Similar tests for cross-section for HONO (Bongartz et al. 1994, Brust et al. 2000, and Stutz et al. 2000) were also performed.

**RC1:**
Line 143. Please add a figure showing an example calibration to the supplemental, stating the literature and observed wavelengths of the neon lamp.
**Reply:** Wavelength calibration is standard spectroscopic practice and we do not think it is crucial to convey this information in this publication. We nevertheless included a new Figure S9 to the supplementary material and refer to it in the main text. We re-numbered the figures in the supplementary material and in the main text accordingly.

**RC1:**
Line 143. "Reflectivity calibration issues" - consider striking "issues" since (I assume) these issues were ultimately resolved.
**Reply:** We agree that the choice of the word issue is not appropriate here. We changed "issues" to "aspects". See also changes concerning the re-numbering of Figure 4 in one of the comments below.

**RC1:**
Line 173. "gas phase titration" with $O_3$?
**Reply:** Correct – information included.

**RC1:**
Line 175. "by determining the corresponding NO yield from HONO numerically from the observed spectrum of the LEDs. The HONO photolysis contribution to NO is less than 5% compared to that of $NO_2$." It is likely that the interferences are underestimated. For example, Andersen et al. (AMT 14, 3071-3085, doi: 10.5194/amt-14-3071-2021) found that "the original BLC used was constructed with a Teflon-like material and appeared to cause an overestimation of $NO_2$ when illuminated". More recently, Gingerysty et al. (Journal of Environmental Sciences 107, 184-193, 2021, doi: 10.1016/j.jes.2020.12.011) found that in their photolytic converter the HONO to NO conversion rate "was larger than predicted from the overlap of the emission and HONO absorption spectra". Further, isoprene photochemistry in the presence of $NO_x$ may lead to formation of PAN and MPAN (as discussed in this paper), which are efficiently converted in the BLC due to it running hot (Reed et al ACP 16,4707, 2016, doi: 10.5194/acp-16-4707-2016). There are also potential interferences from alkenes to consider (Alam et al., AMT 13, 5977, 2020 doi: 10.5194/amt-13-5977-2020).

**Reply:** There are 2 issues concerning HONO interference by a photolytic converter, which were investigated in depth at a European NOx-intercomparison campaign at the observatory Hohenpeißenberg, Germany (10.-20. October 2016):

(a) the spectral interference defined by the HONO absorption spectrum and the LED spectrum; this gives the interference value for the limit of small absorptions near zero.

(b) the saturation effect when the conversion efficiency is going towards 100%.

The spectral interference for the LED used in 2011 was determined by spectral radiometer results of the output of the LEDs and the absorption cross sections of $NO_2$ and HONO. The interference (i.e. photolysis_frequency_HONO / photolysis_frequency_$NO_2$) has been 3.5%, caused by the slightly different peak wavelength near 395 nm. The LED had a conversion efficiency for $NO_2$ of 30% in the experiments in 2011. A 5% interference at 30% conversion efficiency of $NO_2$ with the LED was found in the intercomparison campaign (see figure on the effective HONO interference of a BLC below).

[Figure]

This result was determined by sampling the output of a HONO source, a NOy-Au converter for the HONO reference, and the spectral radiometer results for the LED in comparison with another LED. Since the LED in 2011 had a 3.5% photolysis rate ratio, and not 4% as in the campaign in 2016, the estimated HONO interference for the LED at an $NO_2$ conversion efficiency of 30% would be even lower than 5%.

The published results of Gingerysty et al. are in line with these experimental findings. Gingerysty et al. had an observed ratio for a LED at 395 nm of J_HONO/J_$NO_2$ = 6.0% at a $NO_2$ conversion efficiency of >90%. They also state: "CF_HONO was larger than predicted from the overlap of the emission and HONO absorption spectra."

In conclusion, the sentence

"…for the calibration and a ±5% uncertainty for the $NO_2$ conversion efficiency. The known interference of 5% towards HONO is not corrected in the final dataset and not included in this accuracy estimate."

was included in order to be more specific.

**RC1:**
Line 205. Since nitric oxide was quantified, it would be informative to see those data. Please superimpose on the panels showing $NO_2$ mixing ratios.

**Reply:** We included the NO data in a modified (re-numbered) Figure 2(a) as suggested.

**RC1:**
Line 210. "At 9:15 hrs there is a marked but unexplained change in the data" which is difficult to see in the graph as presented (see comment on Figure 1 below).
**Reply:** The corresponding figure has been split into two parts Figure 2(a) and 2(b) (both re-numbered) to account for this comment (see also third last comment below).

**RC1:**
Line 215. "the return to dark conditions at 15:33 hrs led initially to an unexpected increase of the HONO mixing ratio as recorded by the LOPAP instrument," which would be consistent with the LOPAP having an $NO_3/N_2O_5$ interference, is it not? See also line 340.
**Reply:** We acknowledge this comment by the reviewer and added the following text at in Section 4.2 (Kleffmann et al. 2008 was also added in this context):
"A possible interference of the LOPAP instrument towards $N_2O_5$ and/or $NO_3$, which are both formed in the dark chamber in the presence of $NO_2$ together with an excess of ozone, appears likely. However, laboratory investigations of Kleffmann et al. (2002, 2006, 2008) found no evidence for a cross sensitivity for $NO_3$ or $N_2O_5$ for a two-channel LOPAP instrument. This finding is supported by the IBBCEAS measurement data in our experiments which often show the same trend to elevated HONO concentrations in the dark after the roof was closed."

**RC1:**
Line 219. "Almost all NO was oxidised" - please show the NO data to support this statement (see line 205).
**Reply:** We included the NO data in the modified re-numbered Figure 2(a) as suggested previously and referred to this figure in the text to show the evidence for this statement.

**RC1:**
Line 224. What was the temperature of the chamber?
**Reply:** The temperature was >300 K. We included this information in the main text of the manuscript and also added Figure S8 to the supplementary material, showing the temperature as a function of time on 11 July 2011.

**RC1:**
Line 226. "concentrations .... appeared to stagnate" - this is hard to see (see comment on Figure 1). Consider adding the time of this event to the text.
**Reply:** "(at 15:33 hrs)" was added to the text and the corresponding figure changed.

**RC1:**
Line 243. "Vestiges" - Teflon is permeable to small molecules ($O_2$, NO etc.) - how much $NO_x$ is leaking in from the outside?
**Reply:** The amount of $NO_x$ diffusion into the chamber is negligible. The actual simulation chamber is inside a second teflon bag and is kept at slightly higher pressure (~30 Pa) than ambient pressure.

**RC1:**
Line 243. "The variation of temperature was limited to the natural variability." Are the temperature data shown anywhere?
**Reply:** As mentioned above already, we included the temperature data to the supplementary material in Figure S8.

**RC1:**
Line 246. "rather satisfactory" please be quantitative (especially since you can - see Table 1).
**Reply:** We modified the text in this section and wrote:
"The correlation between the data obtained with IBBCEAS and LOPAP, as represented by the correlation coefficient of r = 0.970 (Table 1), is still rather satisfactory at these low levels. For $NO_2$, however, the correlation is less pronounced on this day as there appears to be a gradual drift between data from the IBBCEAS and CLS instruments as evident through a discrepancy in the slope of ~38%. Even though this is the largest discrepancy observed (see Table 1) the data are all taken at $NO_2$ mixing ratios below 300 pptv, which is close to the $2\sigma$ detection limit of 76 pptv. Approximately one third of the data points are below that limit (see Figure 4), which puts these values into perspective."

**RC1:**
Line 246. "For $NO_2$ there appears to be a slight offset" and a 38% slope error - not exactly "good agreement"
**Reply:** We agree with this observation by the reviewer and rephrased the text (see previous comment). However, we would like to point out that on this particular day all measured mixing ratios are below 300 pptv, and more than one third of the data points in the correlation are below the estimated IBBCEAS $3\sigma$ detection limit for $NO_2$. This does not make the correlation better, but it explains why not the same degree of correlation can be expected between IBBCEAS and CLS. In other publications on broadband CEAS detection of HONO and $NO_2$ the mixing ratios of $NO_2$ at low levels are not even experimentally verified (e.g. Yi et al. 2021) but merely estimated through extrapolation (see also discussion of the first comment).

**RC1:**
Lines 289-294 - determination of $R_{eff}$.  "to determine the reflectivity in the clean chamber instead of using $NO_2$ as calibration gas". The description of how this was accomplished is incomplete. Please expand the text and define $I_{LLO}$ in equation (2) and how exactly (which equation?) $L_{LLO}$ was determined. If this procedure has been described elsewhere, cite the appropriate paper(s).
**Reply:** We included 3 references that are of relevance here: (Ruth et al. 2014, Varma et al. 2009, Ruth and Lynch 2008). We are however not quite sure what else to say here in the manuscript.

$$R_{eff}(\lambda) = 1 - \left( \frac{I(\lambda)}{I_0(\lambda) - I(\lambda)} \, d\varepsilon(\lambda) \right) \qquad (S.\,eq.\,1)$$

If the mixing ratio of, e.g. $NO_2$ is known from a CLS measurement (in an otherwise clean chamber), then $\varepsilon$ is known from the known cross-section spectrum of $NO_2$. By measuring $I_0$ and $I$ and knowing the effective cavity length $d$, the reflectivity spectrum can be calculated (Ruth et al. 2014). Once the reflectivity, $R_{eff}$, is known from an $NO_2$ calibration measurement, the optical loss of the low loss (anti-reflection coated) optic (LLO) can also be measured accurately. This is typically done at the time of the initial calibration of $R_{eff}$ (also see Figures S4-S6). The low loss optic can then be used to determine the reflectivity, using the latest "up-to-date" $I_0$ measurement, typically taken in the morning after extensive overnight flushing, when the chamber is clean. However, during an experiment with sample mixture in the chamber, the LLO is merely used to check for changes in the reflectivity. In the clean chamber in principle

the LLO essentially takes the place of the $NO_2$ calibration gas. In the filled chamber, however, insertion of the LLO cannot be used to independently measure absolute reflectivities anymore but only relative changes of same. With target species (and additional loss) in the chamber, one can only retrieve the absolute reflectivity, if $I_0$ has not changed since its last measurement (in the morning for example). Hence one cannot distinguish between a change of $R_{eff}$ and a change of $I_0$ based on the LLO measurement (compare eq. (2) in the main text). One can however figure out how far the setup has drifted from the initial measurement (typically in the morning).

This is a general problem of open path measurements where $I_0$ cannot easily be established at regular intervals. In closed path setups (extractive instruments) an accurately calibrated LLO can be used for reflectivity calibration because a new $I_0$ spectrum can be created readily every time the reflectivity is to be checked based on eq. (2).

It is recommended to check from time to time how accurate the calibration of the LLO still is. Typical calibration measurements are shown in the supplementary material using $NO_2$ (Figure S5) and the LLO (Figure S6).

We decided to include this information in the supplementary material.

**RC1:**
Lines 305-311 "Data evaluation" and lines 312-317 "Detection limits". These sections describe results and should precede Line 249 "Discussion". In fact, I would move Figure 4 ("Data evaluation") even ahead of Figures 1-3 as it is experimental.
**Reply:** We agree with the reviewer here. This is a good suggestions and we moved Figure 4 to the experimental section and re-numbered other figures accordingly.
The following sentence was included in the experimental section:
"An example of an IBBCEAS extinction spectrum and the corresponding fit of eq (1) to the measured data are shown in Figure 1 (uppermost panel). We will further discuss this figure as well as reflectivity calibration aspects in section 4."

**RC1:**
Line 340. Has the potential interference from $NO_3$ / $N_2O_5$ in the LOPAP instrument been evaluated?
**Reply:** This potential interference has now been mentioned in Section 4.2. Information can be found in Kleffmann et al. 2008.

**RC1:**
Line 347. " In the dark (closed) humidified chamber HONO can still be produced " - but the IBBCEAS did not see this ... which suggests an issue with the LOPAP ...
**Reply:** The potential interference issue has now been mentioned in Section 4.2.

**RC1:**
Line 354/355. "The inlet of the LOPAP instrument is much closer to the chamber wall"
Please do not introduce new experimental details and results in a discussion section (i.e., move this information to the experimental setup section and show the LOPAP in Figure S1).
**Reply:** We disagree with the reviewer on this point. The information on the LOPAP inlet was indeed mentioned in the experimental section (line 159/160). The purpose of the discussion section is to find explanations for the observations and/or interpret same. If an experimental detail is important it is of course appropriate to discuss same here. Furthermore, the LOPAP instrument has been published in detail in Häseler et al. 2009 and Li et al. 2014 and an additional figure in the Supplementary material is thus not warranted.

**RC1:**
Isn't the SAPHIR chamber equipped with one (or more) mixing fan(s)? If there are gradients within the SAPHIR chamber, wouldn't have been observed and discussed in the numerous other papers using SAPHIR chamber data? If not, perhaps it should be discussed in this paper in more detail (typical mixing times etc.).
**Reply:** The gases in the chamber were mixed with a fan during the experiment and therefore inhomogeneities of an extent that would explain this observation can be ruled out. The running of the fan leads to the removal of inhomogeneities on a time scale of two to three minutes. The absence of concentration gradients under well mixed conditions was demonstrated during the intercomparison of $OH/HO_2$ detection instruments (Schlosser et al., 2009) and $NO_3/N_2O_5$ detection instruments (Dorn et al. 2013).

**RC1:**
Line 400. What was the detection limit for MACR? This info is hidden in Figure S7 but should be stated here
**Reply:** This info was not 'hidden in Figure S7' but stated in Section 4.1.4. on detection limits. We moved this information into the abstract and included it in the conclusion: "Methacrolein was also detected at mixing ratios below 5 ppbv with a $2\sigma$ detection limit of 340 pptv in the same integration time."

**RC1:**
"competitive" isn't a good choice of words since the other HONO instruments are all mobile (and this is one is not). It probably suffices to say that the LODs here are lower than those of recently described HONO instruments.
**Reply:** We changed the sentence to: "These detection limits are lower in comparison to those reported in the recent literature." We also eliminated the word competitive at the end of section 4.1.4.

**RC1:**
Figure 1. Please modify the figure so that the time series use the full width of the page - the scatter plots could then be superimposed as insets. In the figure as shown, the time series is crammed into a tiny corner, making it difficult to follow the discussion of the main text.
**Reply:** The corresponding figure has been split into two parts (now Figure 2(a) and 2(b) to account for this comment (see also comment concerning line 210 above).

**RC1:**
The supplemental contains a lot of figures that are not sufficiently described or discussed in the main manuscript. Please expand the text.
**Reply:** We added some explanations concerning Figures S4 and S5, and the specifications of the IBBCEAS setup in comparison with the overview given by Jordan and Osthoff, 2020. All other figures are referred to in the main text and in a specific context – more explanation in the supplementary material is not warranted in our opinion.

---

## Author Comment (AC2)

**RC2:   Reply to anonymous Reviewer #1 (15 Dec 2021)**

Dixneuf et al., present an inter-comparison of nitrous acid by open path IBBCEAS and LOPAP instrument in SAPHIR chamber in 2011. They show a good performance of this cavity enhanced absorption technique in measuring HONO, they also compared the measured $NO_2$ and MACR with CLS and PTR. This paper is well written and I only have the follow comments need to be addressed.

**Reply:** We would like to thank the reviewer for the interesting comments which we addressed as good as possible and reasonable.

**RC2:**

1.      The measurement of MACR by IBBCEAS should also be mentioned in Abstract.

**Reply:** The detection of MACR was already mentioned in the abstract. We included the estimated 2 sigma detection limit. The following sentence:
"Methacrolein was also detected at mixing ratios below 5 ppbv."
was changed into
"Methacrolein (MACR) was also detected at mixing ratios below 5 ppbv with an estimated $2\sigma$ detection limit of 340 pptv for the same integration time."

**RC2:**

2.      With respect to the sensitivity change of IBBCEAS in 11 July, is it possibly caused by the unknown vibration and changed the coupled optical system, that means the effective reflectivity may be decreased largely, maybe you can use the retrieved O4 as a tracer to make it clear. If the Reflectivity changed, the intensity of the spectrum before and after the time point of 09:15 maybe also have a large difference.

**Reply:** We think that the change in noise in the HONO retrieval after 9:15 hrs on July 11 is not caused by a sudden change in the mirror reflectivity. No relative reflectivity changes were detected on the day (see also comment 4 below); furthermore the change in noise should in case of optical misalignments also be observed in the retrieval for $NO_2$ and MACR after 9:15 hrs, which does not seem to be the case.
Moreover, $O_4$ cannot be used as tracer in an open path setup. In IBBCEAS the transmission spectrum without the target species, $I_0(\lambda)$, must be known before the transmission with the target species, $I(\lambda)$, is measured; see eq.(1) in the manuscript. In an open path setup, like in the present case (with e.g. target species HONO, $NO_2$ and MACR), the spectrum $I_0(\lambda)$ is taken in a clean and dry air-filled chamber. The spectrum $I_0$ thus already contains the information on the known $O_4$ absorption bands at 360 and 380 nm, i.e. the bands that are relevant here. Since the spectrum $I(\lambda)$ also contains the same bands due to the $O_4$ concentration remaining constant in good approximation, (provided substantial temperature variations that may impinge on the equilibrium concentration of $O_4$ can be neglected.
Therefore $O_4$ cannot be used as a tracer for the reflectivity. That is also reason why $O_4$ was not need (nor incorporated) in the retrieval of HONO and $NO_2$ mixing ratios.

**RC2:**

3.      Line 285 please provide more details of the calculation of effective reflectivity in the text to make this section easier to follow.

**Reply:** We are not quite sure what else to say here in the manuscript.

$$R_{\text{eff}}(\lambda) = 1 - \left( \frac{I(\lambda)}{I_0(\lambda) - I(\lambda)} \, d\varepsilon(\lambda) \right) \tag{R1}$$

If the mixing ratio of $NO_2$ is known from a CLS measurement (in an otherwise clean chamber) then $\varepsilon$ is known from the known cross-section spectrum of $NO_2$. By measuring $I_0$ and $I$ and knowing the effective cavity length $d$, the reflectivity spectrum can be calculated. This approach is well know from numerous other publications. We included the review (Ruth et al. 2014) to guide the reader to secondary literature concerning this aspect.

**RC2:**
4.      The open path IBBCEAS can calibrate alone by an anti-reflection coated optics, why the authors calibrated again by the CLS NOx and then compared with the result of CLS NOx?

**Reply:** Once the reflectivity, $R_{\text{eff}}$, is known from an $NO_2$ calibration measurement, the optical loss of the low loss (anti-reflection coated) optic (LLO) can also be measured accurately. This is typically done at the time of the initial calibration of $R_{\text{eff}}$. The low loss optic can then be used to determine the reflectivity, using the latest "up-to-date" $I_0$ measurement, typically taken in the morning after extensive overnight flushing, when the chamber is clean. However, during an experiment with sample mixture in the chamber, the LLO is merely used to check for changes in the reflectivity. In the clean chamber in principle the LLO essentially takes the place of the $NO_2$ calibration gas. In the filled chamber, however, insertion of the LLO cannot be used to independently measure absolute reflectivities anymore but only relative changes of same. With target species (and additional loss) in the chamber, one can only retrieve the absolute reflectivity, if $I_0$ has not changed since its last measurement (in the morning for example). Hence one cannot distinguish between a change of $R_{\text{eff}}$ and a change of $I_0$ based on the LLO measurement (compare eq. (2) in the manuscript). One can however figure out how far the setup has drifted from the initial measurement (typically in the morning).
This again is a problem of open path measurements where $I_0$ cannot easily be established at regular intervals. In closed path setups (extractive instruments) an accurately calibrated LLO can be used for reflectivity calibration because a new $I_0$ spectrum can be created easily enough every time the reflectivity is to be checked based on eq. (2).
It is recommended to check from time to time how accurate the calibration of the LLO still is. In the manuscript we showed typical calibration measurements in the supplementary material using $NO_2$ (Figure S5) and the LLO (Figure S6).
We left the text in the manuscript.

**RC2:**
5.      How about the stability of Reflectivity day by day?

**Reply:** $I_0$ was taken in a clean and dry chamber every morning and the reflectivity was measured accordingly. $R_{\text{eff}}$ calibrations using the calibrated low loss optic were taken at around the same time, in the clean and dry chamber. Over the course of the day several measurements using the LLO were taken regularly, to check that no drift were occurring. This was done together with regular stray light measurements regularly (changes mainly with roof opening/closing). On the days on this paper no dramatic changes beyond the errors stated in the publications were observed (compare Figure S5).

**RC2:**
6.      Line 326, "see Table in the AMT…" please rewrite it with more professional form and cite the Reference.

**Reply:** We changed the phrase to

"- see Table 1 in the publication by Jordan and Osthoff, Atmos. Meas. Tech., 13, 273-285, 2020 (doi: 10.5194/amt-13-273-2020)." We also included some extra information concerning this comparison in the supplementary material.